# Collective movement of schooling fish reduces the costs of locomotion in turbulent conditions

**Yangfan Zhang**[1]*, **Hungtang Ko**[2], **Michael A. Calicchia**[3], **Rui Ni**[3], **George V. Lauder**[1]

**1** Museum of Comparative Zoology, Department of Organismic and Evolutionary Biology, Harvard University, Cambridge, Massachusetts, United States of America, **2** Department of Mechanical and Aerospace Engineering, Princeton University, Princeton, New Jersey, United States of America, **3** Department of Mechanical Engineering, Johns Hopkins University, Baltimore, United States of America

* yangfan_zhang@fas.harvard.edu

**Data Availability Statement:** All data and custom code are available at Harvard Dataverse: https://doi.org/10.7910/DVN/CVNLZE.

**Funding:** Funding provided by the National Science Foundation grant 1830881 (GVL), the Office of

## Abstract

The ecological and evolutionary benefits of energy-saving in collective behaviors are rooted in the physical principles and physiological mechanisms underpinning animal locomotion. We propose a turbulence sheltering hypothesis that collective movements of fish schools in turbulent flow can reduce the total energetic cost of locomotion by shielding individuals from the perturbation of chaotic turbulent eddies. We test this hypothesis by quantifying energetics and kinematics in schools of giant danio (*Devario aequipinnatus*) and compared that to solitary individuals swimming under laminar and turbulent conditions over a wide speed range. We discovered that, when swimming at high speeds and high turbulence levels, fish schools reduced their total energy expenditure (TEE, both aerobic and anaerobic energy) by 63% to 79% compared to solitary fish (e.g., 228 versus 48 kj kg$^{-1}$). Solitary individuals spend approximately 22% more kinematic effort (tail beat amplitude•frequency: 1.7 versus 1.4 BL s$^{-1}$) to swim in turbulence at higher speeds than in laminar conditions. Fish schools swimming in turbulence reduced their three-dimensional group volume by 41% to 68% (at higher speeds, approximately 103 versus 33 cm$^3$) and did not alter their kinematic effort compared to laminar conditions. This substantial energy saving highlights that schooling behaviors can mitigate turbulent disturbances by sheltering fish (within schools) from the eddies of sufficient kinetic energy that can disrupt locomotor gaits. Therefore, providing a more desirable internal hydrodynamic environment could be one of the ecological drivers underlying collective behaviors in a dense fluid environment.

## Introduction

Nearly all animal species live with ubiquitous turbulent air or water in nature [1–4]. Hence, turbulent flows affect many aspects of animal biology that are fundamental to lifetime fitness, including dispersal and spawning, the cost of moving for both regional locomotion and long-distance migration, and the dynamics of predator–prey interactions [5]. In particular, chaotic

Naval Research grants N00014-21-1-2661 (GVL), N00014-16-1-2515 (GVL), 00014-22-1-2616 (GVL), and a Postdoctoral Fellowship of the Natural Sciences and Engineering Research Council of Canada (NSERC PDF - 557785 – 2021) followed by a Banting Postdoctoral Fellowship (202309BPF-510048-BNE-295921) of NSERC & CIHR (Canadian Institutes of Health Research) (YZ). The funders had no role in study design, data collection and analysis, decision to publish, or preparation of the manuscript.

**Competing interests:** The authors have declared that no competing interests exist.

**Abbreviations:** COT, cost of transport; EPOC, excess post-exercise oxygen consumption; FFT, fast Fourier transform; PIV, particle image velocimetry; RPM, revolutions per minute; TCOT, total cost of transport; TEE, total energy expenditure.

turbulent flows [6–8] directly subject solitary individuals to unpredictable fluid fields and alter their body kinematics. This challenge is especially formidable for aquatic life in natural channels of rivers and coastal seas [9–12], because water is 50 times more viscous than air [5,13–15] and exerts larger perturbing forces on aquatic vertebrates.

Moving in turbulence is particularly challenging and energetically expensive for solitary fish. Solitary creek chub (*Semotilus atromaculatus*) swimming in turbulence reduced maximum sustained swimming speed (by 22%) because large turbulent eddies (approximately 76% of body length) disrupt the movement trajectories of fish [14]. Also, the cost of locomotion by solitary Atlantic salmon (*Salmo salar*) can increase by approximately 150% in turbulence [16]. Moreover, studies on animal locomotion and turbulence have profound implications for a better understanding of the planetary ecosystem, e.g., turbulence generated by groups of fish can contribute to vertical mixing of the ocean [17–19]. Despite the widespread interest in understanding how fish interact with turbulence [5,14,15,18–27] and the ubiquitous interactions of animals and their turbulent fluid environments, no previous study has investigated the effects of turbulent flow on fish schools. This can be due to the complexity of turbulent flow and the dynamic nature of collective animal motion.

However, could the collective movement of fish schools mitigate the effects of turbulent flow by altering their locomotor characteristics and through the coordination of nearby individuals? Nearly all fish could modify the local fluid environment through vortices shed by their undulatory body motion, and by acting as nearby solid surfaces [28–32]. A school of fish could reduce the intensity and length scale of oncoming turbulent eddies within the school (Fig 1). Hence, we propose a "turbulence sheltering hypothesis" that a fish school can shield individuals within the group from ambient turbulence (Fig 1). If this hypothesis holds, a key prediction is that collective movement reduces the total energy expenditure (TEE) per unit of biomass compared to that of a solitary individual under the same flow conditions. This study focuses on testing this hypothesis experimentally.

Vertebrates use both aerobic and non-aerobic metabolic energy to support their TEE during locomotion. Aerobic metabolism primarily supports energy use at slower and steady locomotion, while glycolytic metabolism supplies faster and unsteady state high-speed movement [33]. Not only do the physiological mechanisms underpinning locomotion shift with speed, but also fluid drag scales as the square of fluid velocity. Hence, increased movement speeds physically require substantially more metabolic energy. The characterization of a locomotor performance curve (TEE as a function of speed) [34] under both turbulent and laminar conditions will test the working hypothesis that fish schools could mitigate the expected increase in the energetic cost of moving in turbulence. To test this hypothesis, we directly quantified the locomotor performance curve for both solitary individuals and schools of 8 giant danio (*Devario aequipinnatus*) across a wide range of speeds from 0.3 to 8 body lengths sec$^{-1}$. We measured whole-animal aerobic energy expenditure (oxidative phosphorylation) during swimming, as well as excess post-exercise $O_2$ consumption (EPOC) to quantify non-aerobic energy expenditure after swimming (high-energy phosphates, $O_2$ stores, and substrate-level phosphorylation) [35–37]. In addition, we simultaneously quantified the kinematics of individual fish and those within schools, and measured three-dimensional school volumes to characterize how fish responded to both laminar and turbulent flow environments.

## Results

### Hydrodynamics of turbulent conditions

Turbulent flows (generated by a passive turbulence grid) exhibited strong fluctuations and chaotic patterns (Fig 2A and 2B) in contrast to laminar flows (generated by a flow

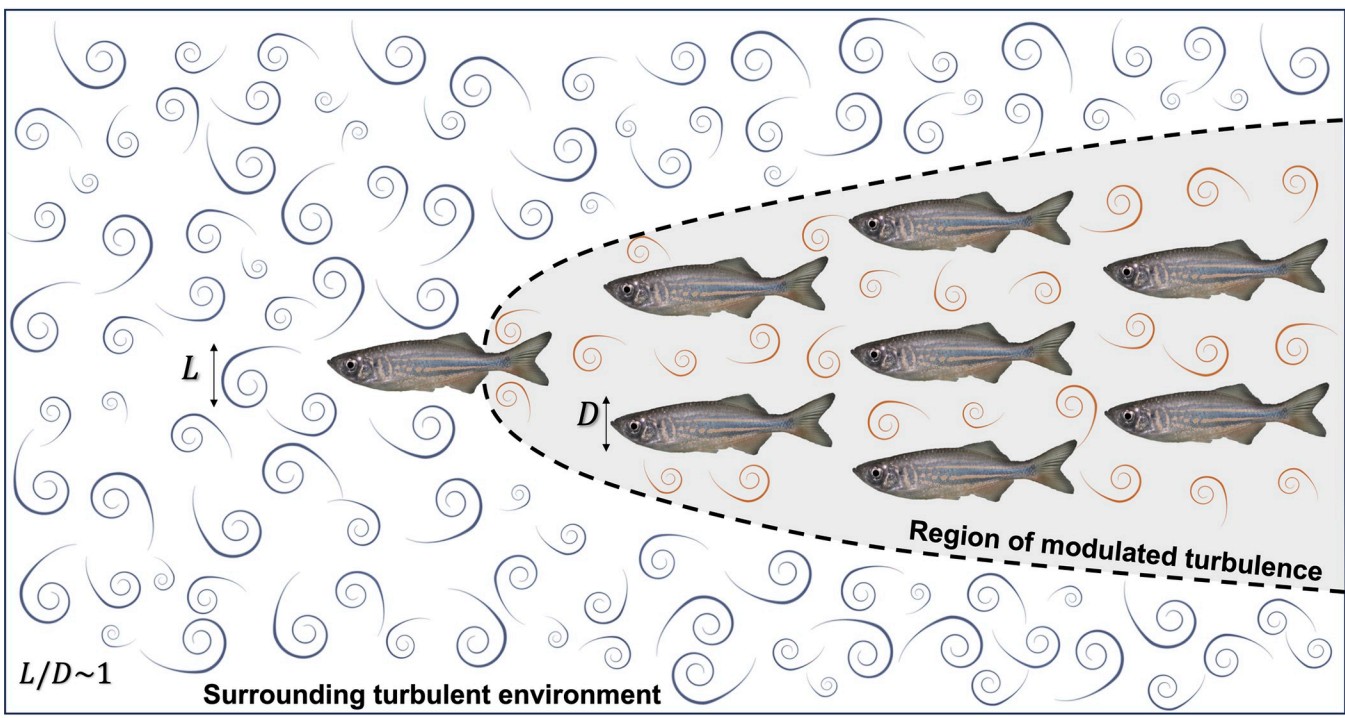

**Fig 1. Illustration of the environmental turbulence sheltering hypothesis.** Schematic diagram of a school of giant danio (*D. aequipinnatus*) swimming in oncoming turbulence where the largest eddies have an integral length scale (*L*) on the same order of magnitude as the body depth (D) of the fish. Fish within the school could benefit from a region of reduced turbulence created within the school as a result of nearby neighbors and undulatory body motion modifying flow within the school compared to free stream oncoming flow. We propose a "turbulence sheltering" hypothesis that fish schools can protect individuals within the group from free-stream turbulence. As a result, we predict fish swimming in turbulence could reduce their locomotor costs by schooling in contrast to swimming alone. The sheltering zone is drawn to start with flows generated by the dorsal and anal fins of the leading fish as these fins generate vortical wakes that could contribute to modifying flow within the school.

straightener). Quantitatively, turbulent testing conditions showed sustainably greater maximum velocity ($F_{1,107}$ = 401.9, $p < 0.001$, Fig 2C), maximum vorticity ($F_{1,107}$ = 167.8, $p < 0.001$, Fig 2D), and maximum (Fig 2E) and sum shear strength ($F_{1,107} \geq 153.7$, $p < 0.001$, Fig 2F). Since turbulence was generated by passing the flow through a passive grid, greater turbulence was reached at higher mean flows; a similar rise of all these parameters in the laminar flow condition was also observed, but with a greatly reduced rate of increase with velocity (Fig 2). In addition, the probability density function (p.d.f) of flow velocity along the fish swimming direction (*u*) and perpendicular direction (*v*) were broadening (Fig 3A and 3B), a clear indication of intensified turbulence as speed increases. Since turbulence fluctuation velocity increased approximately linearly with the mean flow speed (Fig 3C), the resulting turbulence intensity, defined as the ratio between the two, remains nearly constant over the flow speeds studied.

Eddies of various sizes can differentially impact the energetics of fish locomotion. The undulatory motion of fish can respond to eddies smaller than fish size, whereas eddies comparable to the body size may have sufficient energy to change the fish's movement trajectory, which could result in increased energy expenditure. However, turbulence is notorious for its wide spectrum of scales, which can be quantified by the largest (integral scale) and the smallest (Kolmogorov scale), shown as 2 separate lines with the range in between showing the full range (Fig 3D). Based on the framework of turbulence scales and energy cascade, the large eddies at the integral length scale (L) were comparable to the fish body depth (D) (i.e., L/D ~ 1) and were capable of energetically challenging fish locomotion (Figs 1 and 3).

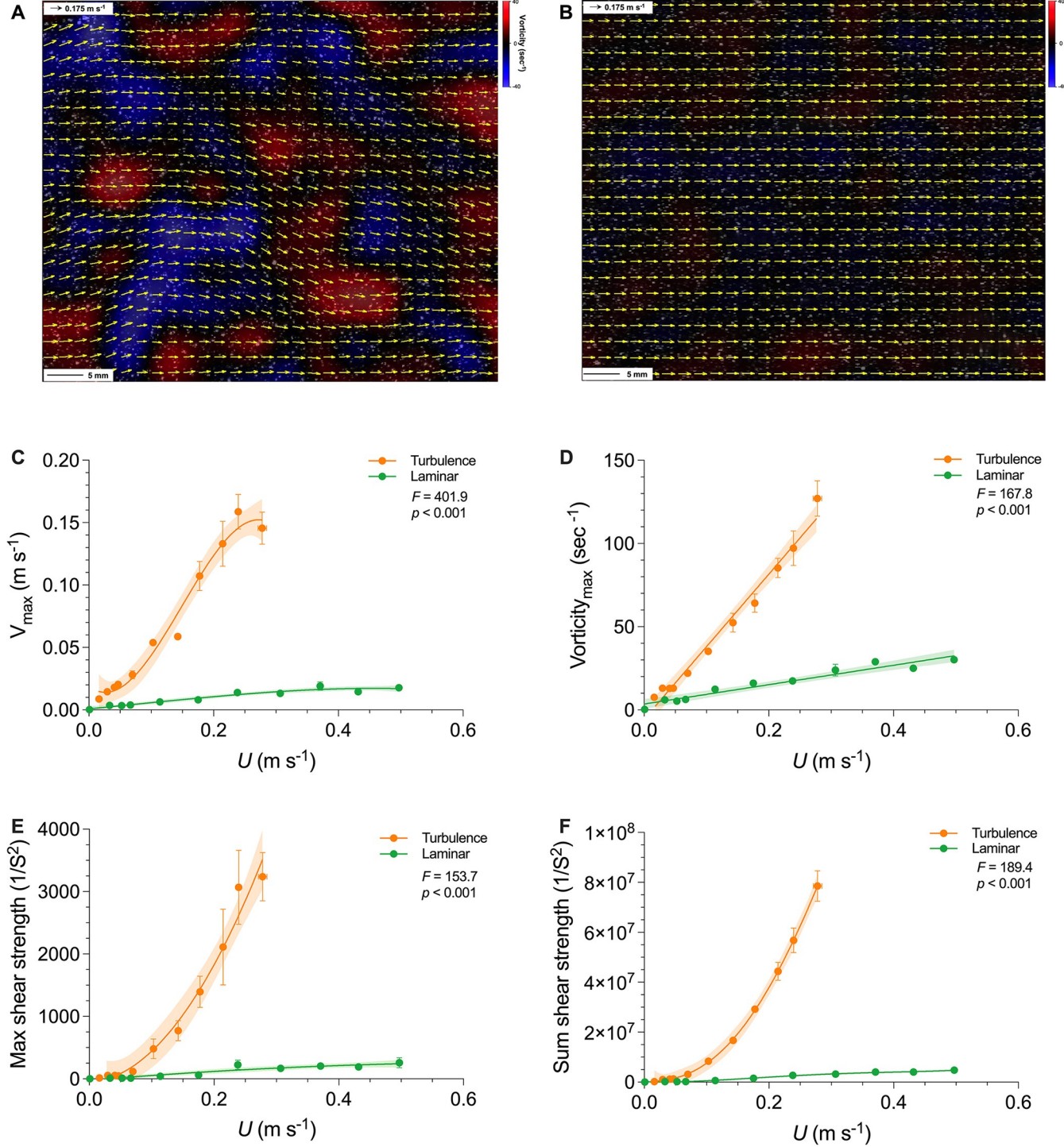

**Fig 2. Characterization of hydrodynamic features in laminar and turbulent flows across a range of speeds used for danio schooling energetic and kinematic measurements.** Representative flow patterns of (**A**) turbulent and (**B**) laminar conditions (both at 18 cm s$^{-1}$ mean flow) in the swim-tunnel respirometer as quantified by particle image velocimetry. Velocity vectors are yellow arrows, and the vorticity field is shown by blue (−40 s$^{-1}$) to red (40 s$^{-1}$) gradient heat maps in the background. The hydrodynamic features of turbulent (orange color) and laminar (green color) flows are characterized by (**C**) maximum vertical velocity, (**D**) maximum vorticity, (**E**) maximum, and (**F**) the sum of shear strength as a function of absolute speed (meter sec$^{-1}$). More detailed flow characteristics are illustrated in the Supplemental materials for characteristics of the turbulent flow generated in the swim-tunnel respirometer (S2 Fig in S1 Text). The statistics in each panel denote the main effect of the flow condition. Shading indicates the 95% confidence interval. Statistical details are available in the statistical analyses section. The underlying data of this figure are in doi.org/10.7910/DVN/CVNLZE.

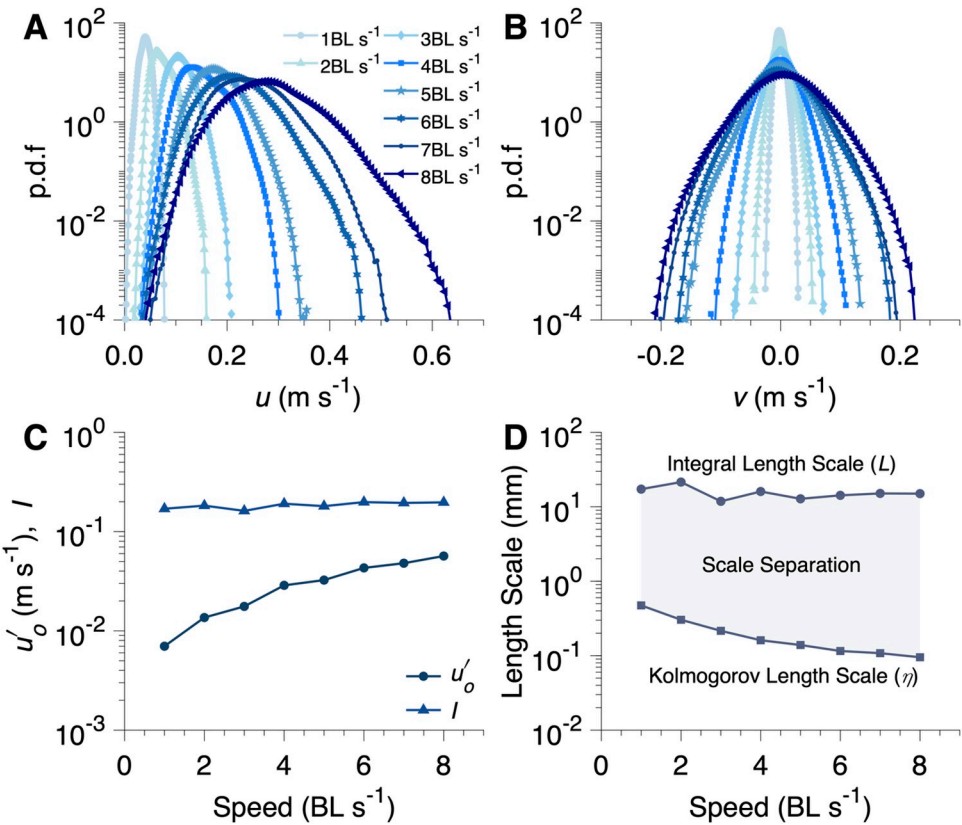

**Fig 3. Analysis of turbulent flow in the respirometer used for testing schooling energetics.** Probability density function (p.d.f) of velocity components **(A)** parallel ($u$) and **(B)** perpendicular ($v$) to the swimming direction for different **swimming speeds** with the passive turbulence grid. As speed increases, the width of the distribution increases, signifying increasing turbulence. **(C)** The turbulence intensity ($I$), the ratio of the standard deviation of velocity to the mean velocity, and fluctuation velocity ($u'_o$) as a function of swimming speed. Fluctuation velocity increases with increasing speed. However, the increase in the fluctuation velocity is proportional to the increase in the speed resulting in a near constant turbulence intensity. (D) Distribution of eddy sizes present for a given swimming speed, from largest (integral length scale) to smallest (Kolmogorov length scale). The eddy size distribution was determined by approximating the energy dissipation rate via computation of the two-dimensional structure functions (see Supplemental material for further information). The largest eddies were roughly the same size as the fish's body depth (L/D = ~1 or 30% of their body length). The underlying data of this figure are in doi.org/10.7910/DVN/ CVNLZE.

## Energetics of collective movement

We discovered that aerobic metabolic rate–speed curves of fish schools and solitary individuals both were concave upward in turbulence across the entire 0.3 to 7 BL s$^{-1}$ speed range (Fig 4). We demonstrated that turbulent flow has resulted in upshifted aerobic metabolic rate–speed curves of solitary individuals across the entire speed range ($F_{1, 69} = 4.45$, $p = 0.0003$) compared to locomotion by fish schools. In particular, the energetic cost of swimming was 32% lower at 6 BL s$^{-1}$ in schools compared to individuals swimming alone in turbulence (719.3 versus 1,055 mg O$_2$ kg$^{-1}$ h$^{-1}$, $F_{1,69} = 4.1$, $p = 0.0015$, Fig 4A). Aerobic energy conservation enabled by schooling dynamics was more pronounced at higher speeds when the energy demands increased exponentially at higher speeds and when aerobic capacity is limited.

Because fish body musculature operating at high frequencies during locomotion at high speed mostly uses white muscle fibers powered in part through glycolysis [38,39], we predict that the schooling dynamics should also conserve non-aerobic energy (estimated by EPOC)

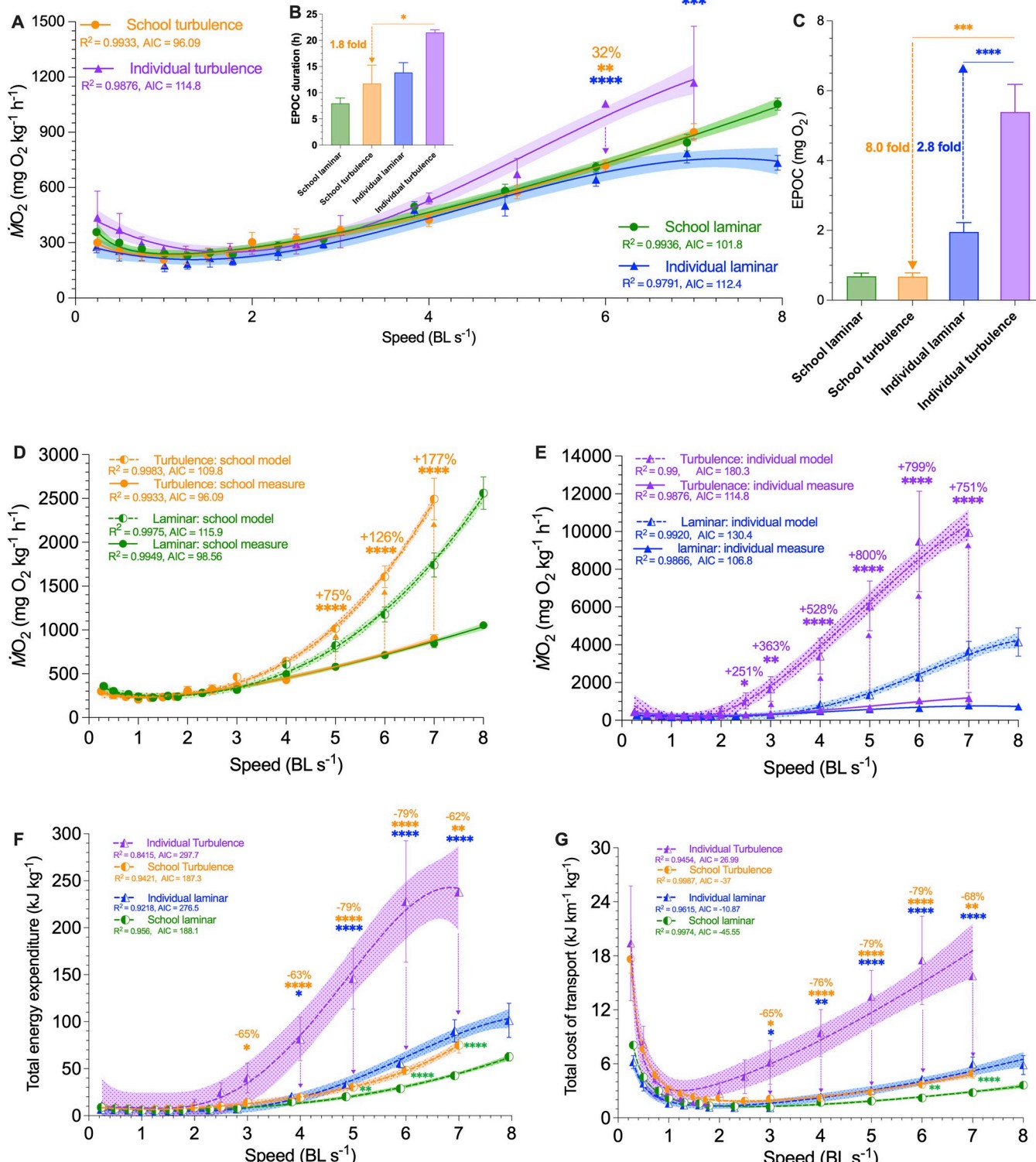

**Fig 4. Measurements of aerobic and non-aerobic locomotor costs for fish schools and solitary fish in laminar and turbulent flow conditions.** The total metabolic costs-speed curve is calculated from a model (dashed line) that integrates the measurements (solid line) of aerobic and non-aerobic locomotion costs (see Supplemental material). (**A**) Comparison of concave upward aerobic metabolic rate ($\dot{M}O_2$)-speed curve over 0.3–8 body length s$^{-1}$ (BL s$^{-1}$) range for fish schools and solitary fish swimming in laminar and turbulent flow conditions. (**B**) Recovery time of EPOC and (**C**) EPOC of fish schools and solitary fish after swimming in laminar and turbulent conditions. Concave upward total $\dot{M}O_2$-speed curve of (**D**) fish schools and (**E**) solitary fish when swimming in laminar and turbulent conditions. (**F**) TEE and (**G**) concave upward TCOT-speed curves for fish schools and solitary fish when swimming in laminar and turbulent

conditions. TEE and the TCOT are calculated using the sum of aerobic and non-aerobic costs. Green color = fish schools in laminar conditions ($n = 5$); blue color = solitary fish in laminar conditions ($n = 5$); orange color = fish schools in turbulence ($n = 4$); purple color = solitary fish in turbulence ($n = 3$). Statistical significance is denoted by asterisk(s). Shading indicates the 95% confidence interval. Statistical details are available in the statistical analyses section. The underlying data of this figure are in doi.org/10.7910/DVN/CVNLZE. EPOC, excess post-exercise oxygen consumption; TCOT, total cost of transport; TEE, total energy expenditure.

and reduce the recovery time when compared with solitary individuals. Indeed, the non-aerobic cost for fish schools to swim through the entire speed range in turbulence was nearly 87% lower than that of solitary individuals (EPOC: 0.68 versus 5.4 mg $O_2$, $t = 7.0$, $p = 0.0005$, Fig 4C). Fish that swam in schools recovered 82% faster than solitary individuals (EPOC duration: 11.8 versus 21.5 h, $t = 2.3$, $p = 0.035$).

As a result, both total energetic expenditure (TEE) and total cost of transport (TCOT) of fish schools were 62% to 79% lower than that of solitary individuals swimming in turbulence over the 3 to 7 BL s$^{-1}$ range (e.g., TEE: 228 versus 48 kJ kg$^{-1}$; TCOT: 17.5 versus 3.7 kJ km$^{-1}$ kg$^{-1}$, Fig 4G). Non-aerobic costs contributed 72% to 83% of total energy consumption in solitary fish, whereas the non-aerobic contribution was only 20% to 40% in fish schools (S1 Table). If non-aerobic locomotor costs were not accounted for, 75% to 177% of locomotor energy expenditure in solitary individuals (Fig 4D), and 251% to 800% of the energy expenditure in fish schools (Fig 4E) would have been overlooked.

Despite the substantial energy saving enabled by fish schooling in turbulence compared to swimming in the same turbulent environment alone, fish schools did not completely eliminate the effects of turbulence on locomotor cost. After accounting for both aerobic and non-aerobic energy costs, we discovered that the TEE of fish schools swimming in turbulence was 51% to 76% higher (e.g., 75 versus 43 kJ kg$^{-1}$, $F_{1,7} = 54.3$, $p \leq 0.0072$, Fig 4F) than the cost for fish schools swimming under laminar hydrodynamic conditions over the speed range of 5 to 7 BL s$^{-1}$, and TCOT of fish schools was 68% and 76% (3.7 versus 2.2 kJ km$^{-1}$ kg$^{-1}$ and 5.0 versus 2.8 kJ km$^{-1}$ kg$^{-1}$) higher in turbulence than in laminar flow ($F_{1,91} \geq 36.2$, $p \leq 0.007$) at 6 and 7 BL s$^{-1}$, respectively (Fig 4G). The proportional increase in the total energetic cost of swimming in turbulence was again mostly from non-aerobic energy production. Nevertheless, the magnitude of the increased total costs of swimming in turbulence was only a fraction (9% to 24%) of the costs for solitary individuals. Solitary individuals spent 190% to 342% higher total energy (e.g., 228 versus 56 kJ kg$^{-1}$) swimming in turbulence across the speed range of 2.5 to 7 BL s$^{-1}$ compared to locomotion in laminar fluid conditions (Fig 4F and 4G).

## Kinematics

Using high-speed videography, we discovered that, as speeds increased to >1.7 BL s$^{-1}$, school volume in turbulence became smaller, and the 3D convex hull representing school volume was reduced by 74% as individuals within the school swim in closer proximity (129 versus 34 cm$^3$; $F_{12,143} = 11.74$, $p < 0.0001$). Over the speed range of 1.7 to 7 BL s$^{-1}$, fish schools became denser compared to the streamlined school structure when fish schools swam in laminar flows of the same speed. The 3D convex hull volume of fish schools swimming in turbulent conditions was 40% to 68% lower than that for schools swimming in laminar conditions at the same speeds (e.g., 103 versus 33 cm$^3$; $t_{1,8} \geq 2.147$, $p \leq 0.032$, Fig 5).

Kinematics of individual fish (Tail beat frequency, $f_{tail}$; tail beat amplitude, Amp$_{tail}$; $f_{tail} \cdot$Amp$_{tail}$) within schools swimming in turbulence was not different from that of fish schools swimming in laminar flow conditions (Fig 6F, MANOVA: $F_{1,25} \leq 0.028$, $p \geq 0.868$). However, solitary fish spend up to 22% more effort (estimated as tail beat frequency times amplitude, $f_{tail} \cdot$Amp$_{tail}$: 1.7 versus 1.4 BL s$^{-1}$) swimming in turbulence than for laminar flow locomotion

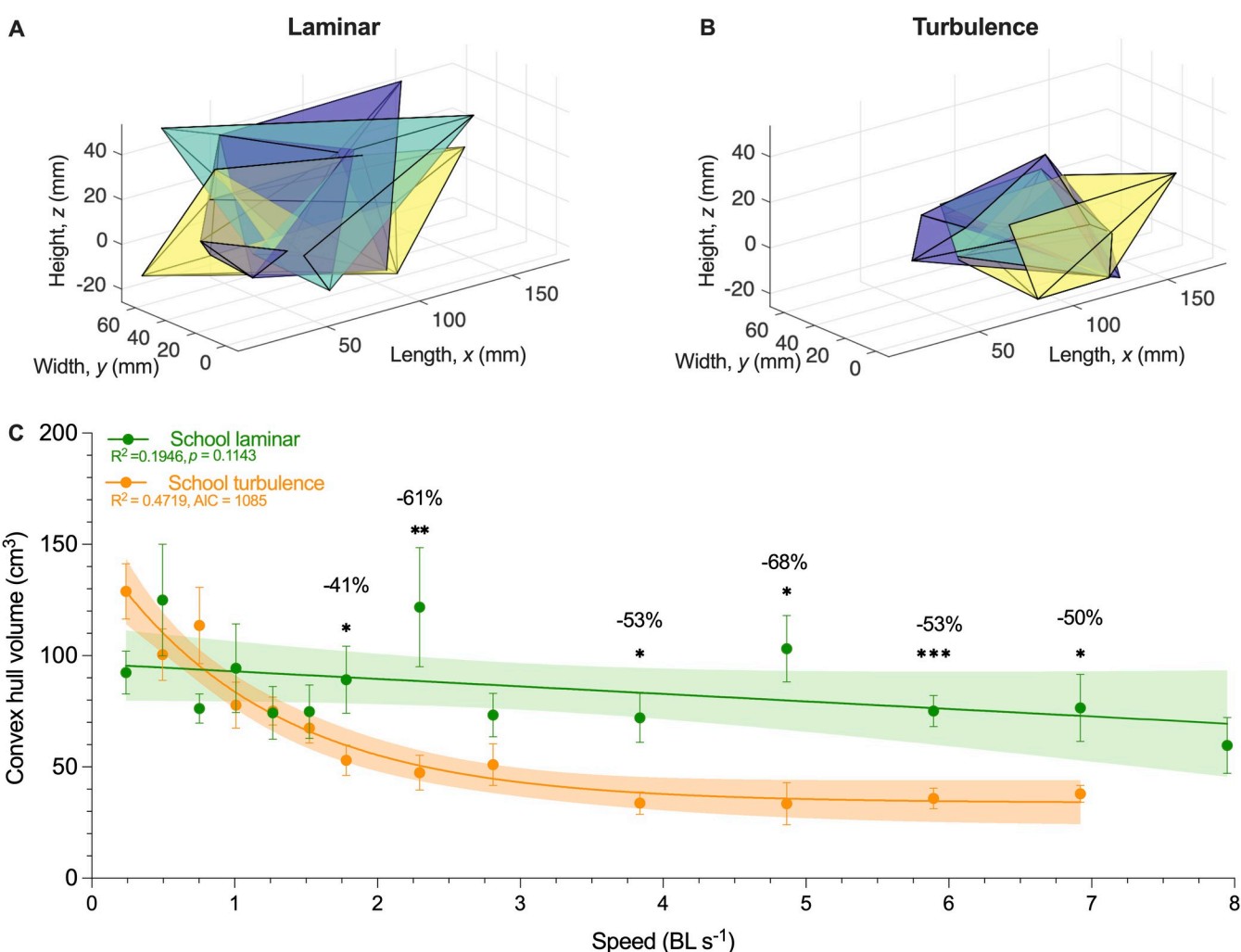

**Fig 5. Characterization of fish school three-dimensional volume in turbulent and laminar flow conditions.** Representative three dimensional (3D) convex hull volume of fish schools in (**A**) laminar and (**B**) turbulent flow conditions. (**C**) 3D convex hull volume as a function of speed for fish schools swimming in laminar ($n = 9$ snapshots per speed increment) and turbulent ($n = 12$ snapshots per speed increment) flow conditions. Different colors of convex hulls in one panel correspond to different sampling points of the same school at the same speed. Statistical significance is denoted by asterisk(s). Shading indicates the 95% confidence interval. Statistical details are available in the statistical analyses section. The underlying data of this figure are in doi.org/10.7910/DVN/CVNLZE.

(Fig 6E, MANOVA: $F_{1,28} \geq 1.167$, $p \leq 0.006$). Solitary fish increased $f_{tail}$ at lower speeds (ANOVA: $F_{1,575} \geq 4.67$, $p \leq 0.031$, Fig 6A) and also reduced $Amp_{tail}$ (ANOVA: $F_{1,639} \geq 26.35$, $p < 0.001$, Fig 6C). As a result, swimming effort remained the same at lower speeds when compared to solitary individuals swimming in laminar flows, as indicated by both kinematics ($f_{tail} \bullet Amp_{tail}$) (MANOVA: $F_{1,28} = 1.167$, $p = 0.281$) and energetics (Fig 6F, ANOVA: $F_{1,71} \leq 1.23$, $p \geq 0.96$). As speed increased in turbulence, solitary fish increased $Amp_{tail}$ by 26% (0.18 versus 0.14 BL; ANOVA: $F_{14,639} = 3.69$, $p < 0.001$; Fig 6C), and there was no further increase in $f_{tail}$ (Fig 6A). Locomotor effort ($f_{tail} \bullet Amp_{tail}$) increased with speed at a greater rate for individuals in turbulence compared to individuals swimming under laminar conditions over the range of speeds tested (ANOVA: $F_{7,536} = 173.14$ $p \leq 0.006$, Fig 6E).

Because of the undulatory locomotion of giant danio, we examined the oscillation at the nostril region of the fish head (at 6 BL s$^{-1}$, a challenging speed for fish while not showing extensive burst-&-gliding swimming gait) to understand the effect of turbulence on the solitary

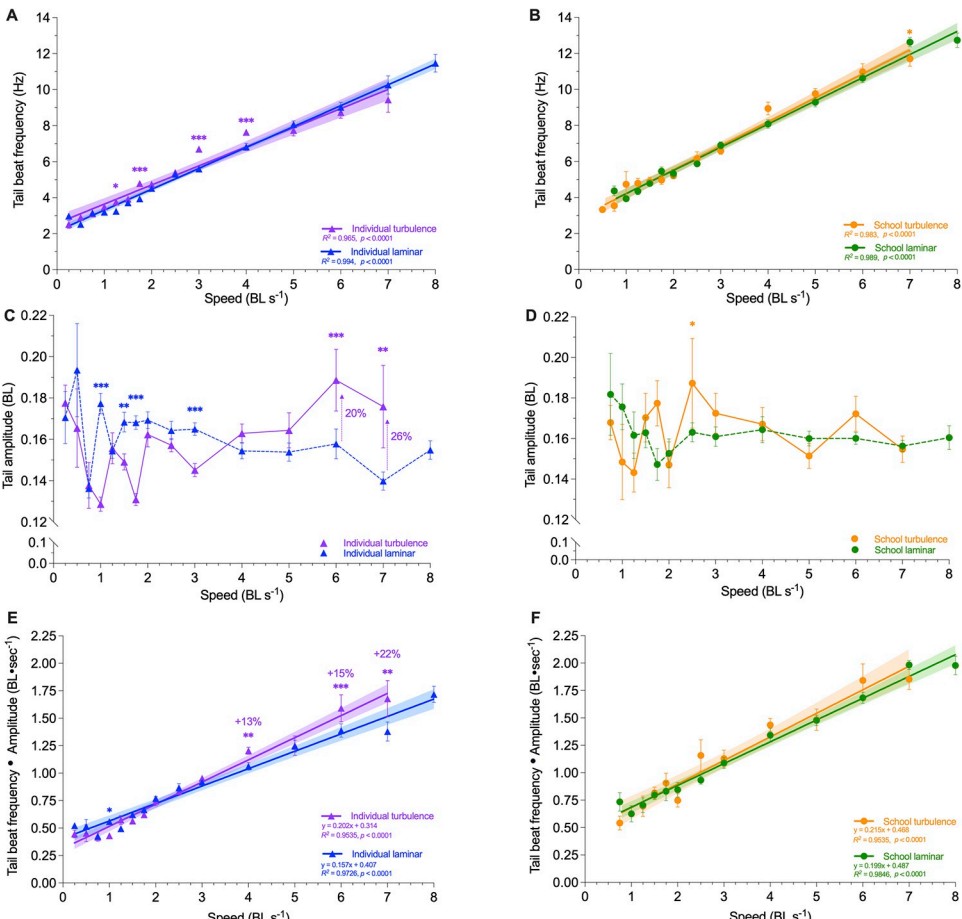

**Fig 6. Kinematic data on individual fish within the school during locomotion in laminar and turbulent flow conditions.** Tail beat frequency ($f_{\text{tail}}$) of (**A**) solitary fish and (**B**) fish schools across 0.3–8 body length s$^{-1}$ (BL s$^{-1}$) in laminar and turbulent flow conditions. Tail beat amplitude (Amp$_{\text{tail}}$) of (**C**) solitary fish and (**D**) fish schools across 0.3–8 body length s$^{-1}$ (BL s$^{-1}$) in laminar and turbulent flow conditions. An estimate of swimming effort, $F_{\text{tail}} \bullet$ Amp$_{\text{tail}}$, of (**E**) solitary fish and (**F**) fish schools across 0.3–8 body length s$^{-1}$ (BL s$^{-1}$) in laminar and turbulent flow conditions. Green color = fish schools in laminar conditions ($n = 295$–379 sequences); blue color = solitary fish in laminar conditions ($n = 351$–416 sequences); orange color = fish schools in turbulence ($n = 104$–146 sequences); purple color = solitary fish in turbulence ($n = 220$–258 sequences). Statistical significance is denoted by asterisk(s). Shading indicates the 95% confidence interval. Statistical details are available in the statistical analyses section. The underlying data of this figure are in doi.org/10.7910/DVN/CVNLZE.

individual and how fish schools provide a hydrodynamic shelter. Solitary individuals and fish schools both showed a nearly consistent rhythmic head oscillation (blue and green waves, Fig 7A). However, solitary individuals swimming in the turbulence showed an irregular head oscillation of varying frequency (purple waves, Fig 7A). When fish schools swam in turbulent conditions, the head oscillation of fish within the school was more regular and rhythmic, a feature more closely resembling that of fish schools swimming in laminar conditions. Indeed, the power spectrum of the head oscillation frequency of fish schools in turbulence was mostly within the range of 10.2 to 11.7 Hz (peak-to-peak in the power spectrum, Fig 7B). The values were within the range when fish schools swam in laminar conditions (peak-to-peak in the power spectrum: 8.3 to 12.7 Hz, Fig 7B). The wide distribution of the power spectrum of the fish schools in both flow conditions was quite a contrast to the single peak power spectrum of the solitary fish swam in laminar (9.2 Hz, Fig 7B) and turbulent conditions (7.3 Hz, Fig 7B).

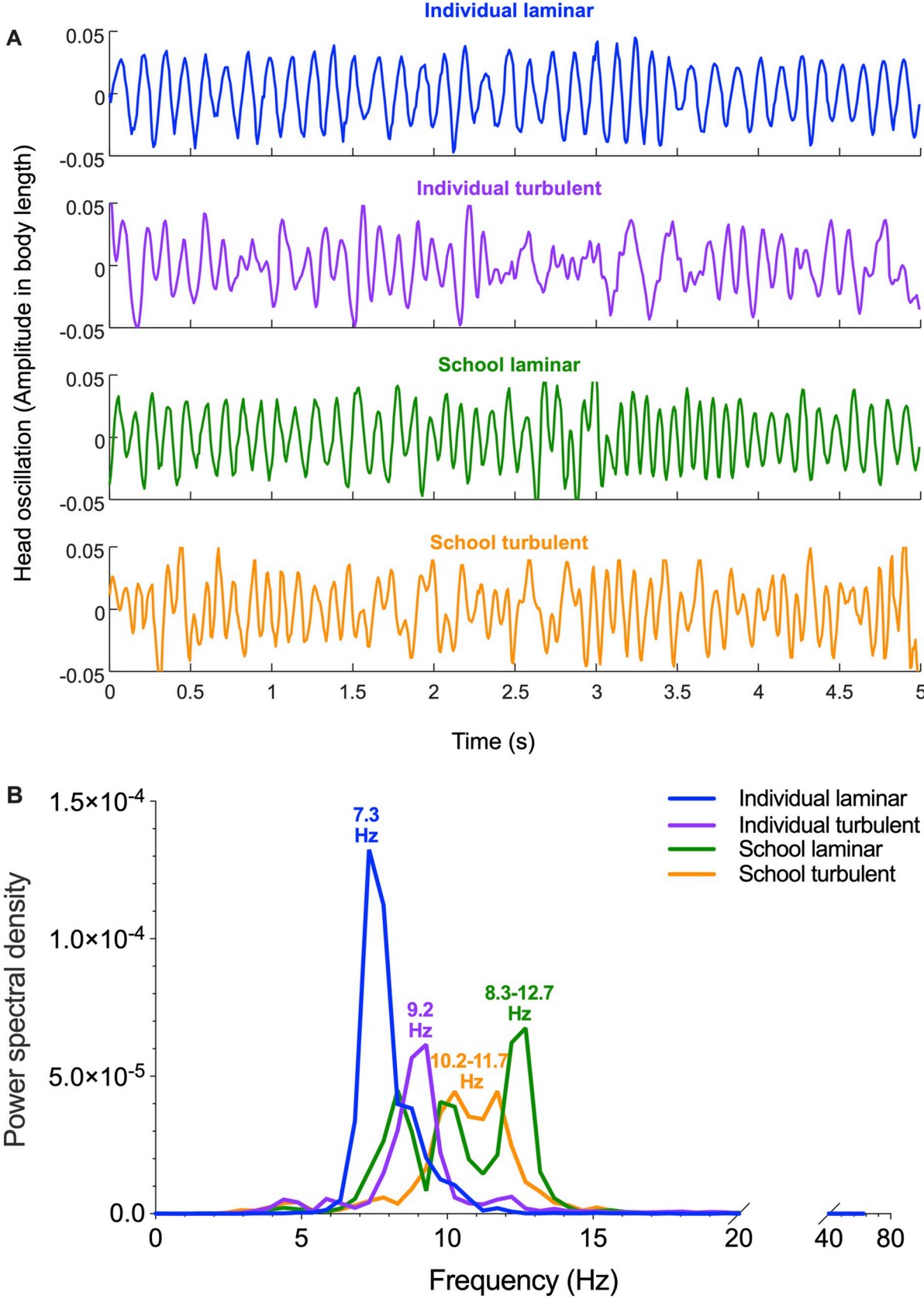

**Fig 7. Responses of schooling fish to the fluid dynamic environment within fish schools under laminar and turbulent conditions.** (**A**) Plots of fish head lateral oscillation through time (blue: individual laminar, purple: individual turbulence, green: school laminar, orange: school turbulence) in water velocity of 6 body length s$^{-1}$ (BL s$^{-1}$). (**B**) Power spectral analysis of head oscillation of an individual in laminar flow (blue), an individual in turbulent flow (purple), fish schools in laminar (green), and fish schools in turbulent flow (orange) swimming at 6 BL s$^{-1}$. The head oscillation frequency peak is denoted for each group. The underlying data of this figure are in doi.org/10.7910/DVN/CVNLZE.

Solitary individuals in turbulence showed a numerically higher frequency peak of head oscillation (9.2 Hz) than that of solitary individuals in laminar conditions (7.3 Hz, Fig 7B), another line of evidence suggesting increased kinematic effort when solitary fish swim in turbulent conditions.

## Discussion

Despite the ubiquity of vertebrates moving in turbulent flow environments, we know remarkably little about the energetic costs of moving in turbulence as a collective group compared to moving in the same conditions as a solitary individual. Hence, we integrate three lines of evidence (energetics, individual kinematics, and schooling dynamics) and compare them between fish schools and solitary individuals over the same speed range in turbulence compared to laminar (control) conditions.

We first discovered that the collective movement of fish schools substantially reduces the effects of turbulence by downshifting the locomotor performance curve at higher speeds (e.g., >50% $U_{crit}$), compared to when solitary individuals swim in turbulence. Fish schools in turbulence expend up to 79% less energy than solitary fish under turbulent conditions (228 versus 48 kJ kg$^{-1}$; Fig 4F). One of the essential mechanisms by which fish schools reduce turbulent disturbances is by collectively swimming in up to a 68% tighter schooling formation compared to laminar conditions (103 versus 33 cm$^3$). As a result, fish swimming within schools were sheltered from the turbulent eddies and showed no difference in their swimming kinematics regardless of whether the fish schools swim in turbulent or laminar conditions. Our results support the turbulence sheltering hypothesis.

We also discovered that one of the key reasons for higher locomotor costs when solitary individuals swim in turbulence was the increase in tail beat amplitude (Amp$_{tail}$) at higher speeds (Fig 6C), which increased the kinematic effort of swimming as estimated by tail beat frequency multiplied amplitude ($f_{tail}$•Amp$_{tail}$) (Fig 6E and 6F). As a result, the TEE and total cost of transport for solitary individuals swimming in turbulent flow, including both the aerobic and non-aerobic energy contributions, are approximately 261% higher compared to when solitary individuals swim in laminar conditions (e.g., 228 versus 56 kJ kg$^{-1}$ Fig 4F and 4G).

Therefore, collective movement provides effective turbulence sheltering, not only mitigating the kinematic responses needed in turbulent flow, but also providing a large energetic advantage by downshifting the portion of the locomotor performance curve at the top approximately 50% of the tested speed range. We highlight 3 key considerations regarding collective movement in turbulent flows: (1) How does collective movement reduce the turbulent disturbance on locomotor energetics? (2) How do body kinematic patterns act as a linchpin between fluid dynamic and energetic effects? (3) How does the hydrodynamic scale of turbulence relative to fish size matter for broader considerations of fish movement ecology?

### (1) Schooling dynamics reduces the effects of turbulence

Fish schools (i.e., giant danio) are effective at reducing the energetic costs of swimming in turbulence. By studying both aerobic and non-aerobic locomotor costs, we discovered that most

of the energy saving stems from the reduced use of non-aerobic energy (S1 and S2 Tables in S1 Text) and related the cost of movement to the gait of locomotion. Collectively, over the entire range of swimming up to maximum and sustained speeds, schooling dynamics effectively dampens the additional metabolic costs of moving in turbulent flow by 74% (the integral area under the TEE curve of schools versus individuals in turbulence: 165.5 versus 633.3 kj kg$^{-1}$). Fish schooling is also highly effective at reducing the perturbation of turbulent eddies on fish kinematics within the school, and fish within a school in turbulence move similarly to fish swimming in laminar flow. Solitary individuals increased tail beat frequency ($f_{tail}$) but reduced tail beat amplitude (Amp$_{tail}$) to compensate for turbulent disturbances at the lower speeds ($\leq$43% critical swimming speed, $U_{crit}$, maximum sustained swimming speed [35,40]), which yielded the same kinematic effort (estimated as $f_{tail}$•Amp$_{tail}$) and energetic cost as swimming under laminar flow conditions. As water velocity increased to $\geq$86% of $U_{crit}$, solitary individuals can no longer increase $f_{tail}$ to mitigate the effects of turbulence but increased Amp$_{tail}$ to mitigate the turbulent disturbances. This increased kinematic effort is reflected in substantially higher energetic cost. The TEE and TCOT of solitary individuals were 188% to 378% higher than that of fish schools at higher speeds, per kilogram of biomass (e.g., TEE: 228 versus 48 kJ kg$^{-1}$; TCOT: 17.5 versus 3.7 kJ km$^{-1}$ kg$^{-1}$). Specifically, the aerobic cost of swimming at higher speeds was also 47% higher in solitary individuals compared to that of fish schools swimming in turbulence which generated an 87% lower excess post-exercise oxygen consumption (EPOC: 0.68 versus 5.4 mg O$_2$, including the use of high-energy phosphate stores and glycolytic energy contributions) than that of a solitary individual.

Bioenergetics is critical to understanding the cost of behaviors [41–43] and allows direct energetic quantification to answer fundamental questions including "How much does a behavior cost?" and "How do altering environmental conditions affect this cost?". Energetic measurements are particularly useful in evaluating hypotheses involving locomotion occurring over a range of movement speeds and investigating the effects of turbulence. One kinematic model for estimating the cost of fish swimming uses tail beat frequency multiplied by tail beat amplitude ($f_{tail}$•Amp$_{tail}$) as suggested by Webb (1973) [44]. This approach suggests that energy saving by schooling danio in turbulence is approximately 22% (tail beat amplitude•frequency: 1.7 versus 1.4 BL s$^{-1}$), whereas direct measurement of energy expenditure shows a total energy saving of approximately 79% (e.g., 228 versus 48 kj kg$^{-1}$). However, a second kinematic model suggests that thrust power scales as ($f_{tail}$•Amp$_{tail}$)$^3$ [45]. Thus, the power needed for an individual fish to swim in turbulence is (1.22)$^3$ times greater than that of schooling fish, which represents an 81% increase in thrust force and closely matches the energetic measurements. Nevertheless, both kinematic models are highly simplified and do not account for individual interactions with fluids within a school under laminar and turbulent conditions which can greatly affect thrust generation as swimming speed changes.

Kinematic analyses that use snapshots of body motion of individuals to quantify biomechanical effort, do not necessarily reflect total energy use, particularly when sampled intermittently during an incremental speed test. Our energetic measurements detected that turbulent flow increased the total cost of locomotion of fish schools by 51% to 76% at higher speeds compared to when fish schools swim in laminar flow conditions (e.g., 75 versus 42 kJ kg$^{-1}$), whereas fish schools swimming in turbulent and laminar conditions showed no detectable difference in tail kinematics. Although understanding the biomechanics of locomotion and movement are essential adjuncts to studying locomotor energetics, the cost of movement for any animal is also governed by the underlying physiological mechanisms relating to the cardiorespiratory system, circulatory system, musculature and metabolic pathways that generate the ATP needed for movement and sustain life [46–49]. We thus advocate here for integrated studies of kinematics and energetics, and caution that kinematic studies alone may not

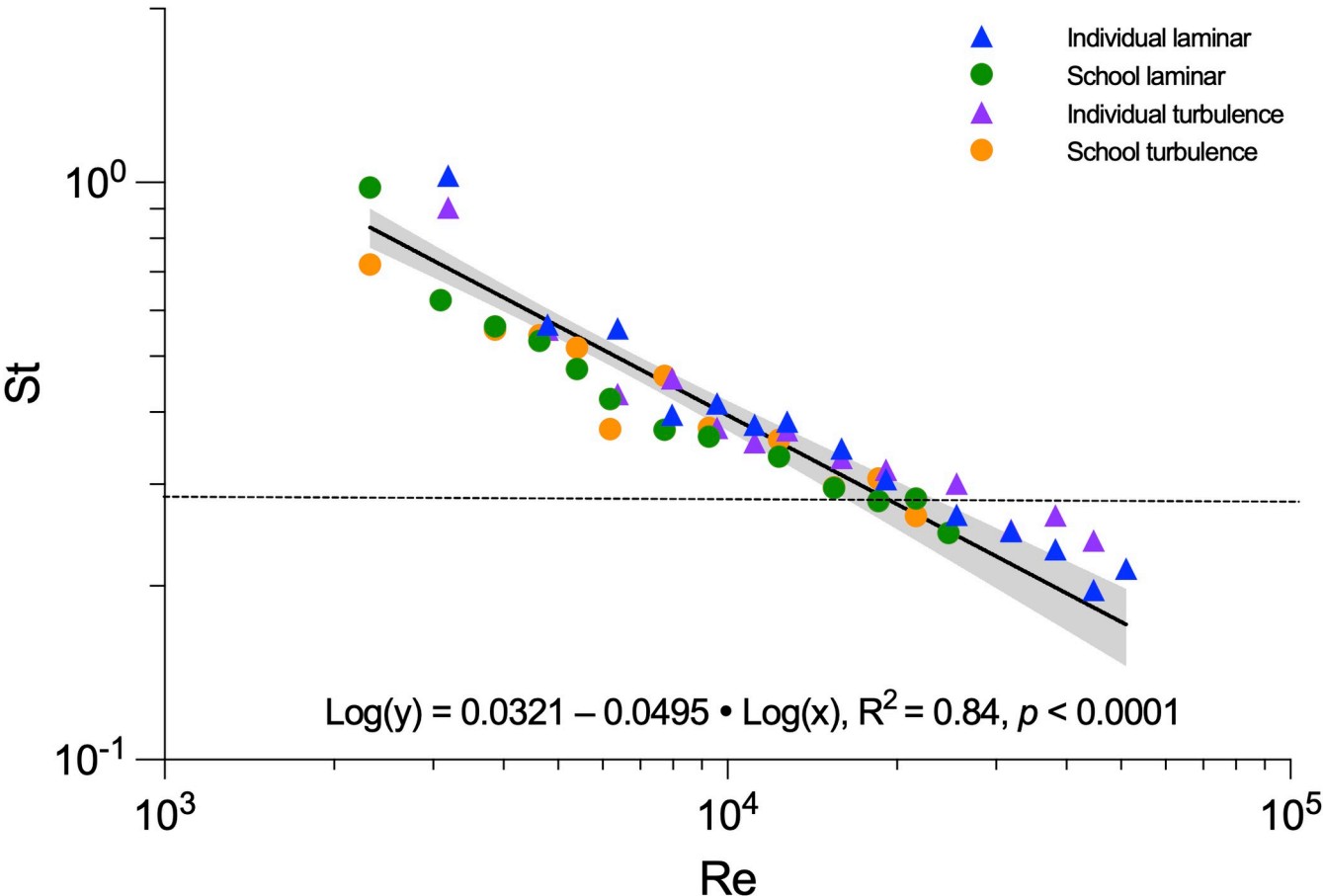

**Fig 8. Relationship between two key dimensionless hydrodynamic parameters (Strouhal number and Reynolds number) for fish swimming in schools and alone under both laminar and turbulent conditions.** Locomotion follows a generally linear relationship ($R^2 = 0.84$, $p<0.0001$). The horizontal dashed line provides the hypothesized relationship of St and Re in the turbulent flow following [45]. Shading indicates the 95% confidence interval. The St number at the lowest speed, when fish exhibited unsteady turning behaviour, is excluded from the analysis to prevent bias in the scaling relationship of directional oscillatory propulsion. The underlying data of this Figure are in doi.org/10.7910/DVN/CVNLZE.

reflect actual levels of energy use, especially in collective groups where complex hydrodynamic interactions change as swimming speed increases.

## (2) Kinematics and schooling dynamics in turbulence

To better understand the physical scaling relationships between fish kinematics and fluid dynamics, we characterized the Strouhal number (St, dimensionless undulatory propulsive effort at a movement speed) [50,51] and Reynolds number (Re, dimensionless ratio between inertial and viscous forces) [13]. Regardless of whether solitary individuals or fish schools swim in laminar or turbulent conditions, the St ranged from 0.25 to 0.35 (Fig 8). The general relationship of St and Re is independent of added turbulence and whether or not fish swim within a school. The log-log linear relationship of Re and St falls in the vicinity of a general scaling hypothesis for aquatic undulatory locomotion (Fig 8) [45]. However, even in turbulent flows giant danios maintain the linear relationship between St and Re instead of transitioning to a different relationship as proposed by a previous scaling hypothesis [45]. It is possible that, if the Re of locomotion is increased beyond Re $> 10^5$ (e.g., larger species moving at faster speeds), the relationship of Re and St could change, but our data do not support the previously

suggested scaling of locomotor St and Re in turbulence. Future laboratory studies are necessary to better inform the scaling relationship of Re and St across a wide range of swimming velocities in ubiquitous turbulent conditions.

Since our data collapse onto a single scaling relationship (Fig 8), how can we explain the difference in metabolic rates between turbulent and laminar conditions? An oscillatory body wave typically starts from the head of the fish. We tracked the oscillation at the nostril region of the fish head (oscillatory amplitude as a function of time) to examine the fluid conditions surrounding the fish. We reason that locations within a school can reduce the turbulent disturbances that would otherwise increase locomotor costs for solitary fish. Solitary individuals and fish schools under laminar conditions show a regular pattern of head oscillation frequency and near-constant oscillation amplitude (Fig 7A). However, the head oscillations of solitary individuals in turbulent flows are irregular and of varying frequency (Fig 7A). Individuals within a school swimming in turbulence share a similar pattern of head oscillation as that of solitary individuals swimming in laminar flow conditions, with more rhythmical head oscillations than those under turbulent conditions (Fig 7A). The benefit of swimming within a school is most likely the result of the tighter schooling formation as fish schooling volumes are reduced in turbulent flows. Indeed, the power spectral analyses of the head oscillation revealed that fish schools swimming in either laminar or turbulent conditions shared an overlapped distribution in the head oscillation frequency (Fig 7B). Moreover, smaller inter-individual distances allow the myriad of hydrodynamic mechanisms that are associated with reduced cost of locomotion to become effective, including fish swimming side-by-side, in front and behind other fish, and in the reduced velocity zone behind two fish [32,34,52–54]. The multiple possibilities of energetically beneficial fish schooling formations underpin the wider frequency distribution of the head oscillation of fish schools in comparison to solitary individuals (Fig 7B). These results suggest that fish schools function as effective "shelters" that enhance hydrodynamic mechanisms that reduce locomotor cost.

As a topic for future investigation, we suggest that the fish schools could alter the size scale of turbulent eddies within the school and thus reduce the impact that environmental perturbations have on the cost of individuals swimming. Fish schools could function as band-pass filters because of their undulatory body motion and the proximity of individuals within the school to each other. This collective undulatory movement could create a more predictable flow field within the group that reduces kinematic efforts and swimming costs compared to individuals swimming alone in turbulence.

Measuring flow conditions within a school would provide a more direct and detailed understanding of how turbulent eddies within fish schools are modified in comparison to the free-stream turbulent flow field. However, this is currently a considerable experimental and technical challenge. Even when using multiple laser light sheets or volumetric approaches to illuminate flow within a school, the bodies of fish in a dense school inevitably cast shadows and hinder the resolution of imaging turbulent flow within the school. In the absence of direct measurements for within-school flow fields, kinematic and energetic data provide the best direct and available evidence for the turbulence sheltering hypothesis.

## (3) Turbulence length scale and fish ecology

Turbulent disturbances on animal movement are multifaceted and context-specific. Turbulent flows are chaotic and unpredictable by fish, and contain energy over a wide spectral range (Figs 2 and 3), which differs from a Kármán vortex wake where a solitary fish can interact with a regular and predictable pattern of oncoming vortices [55]. Fish swimming in a Kármán vortex street can save energy by tuning their body dynamics to interact with oncoming vortices,

and greatly reduce their muscle activity and energetic cost [3,56]. When fish are exposed to the true chaotic turbulent flow with a length scale on the same order of magnitude of body size, as in the experiments presented here, the energetic cost of locomotion greatly increases.

Demonstrating the reduction in the total cost of locomotion in turbulence by group dynamics in aquatic vertebrates has direct implications for the movement ecology of migratory species. For example, a "feast-or-famine" life history is common to many migratory species. Food availability can be scarce and fluctuate during the migration journey [57]. Migratory (fish) species typically accumulate energy and nutrients during the feeding season prior to undertaking a long migration. As a result, migratory species often rely on a finite number of onboard stores of metabolic substrates to fuel migration. We demonstrated here that the total cost of transport when fish schools move through turbulence is substantially decreased. Thus, fish migrating in a collective group, as many fish species do, should travel a longer distance using the same amount of energy.

Fish encounter turbulence not only in natural environments as a result of rapid stream flows, bottom topography, or obstacles in the water, but also under conditions where human-designed structures such as dams or fish passage structures create turbulence [58–60]. A key issue in considering how fish must contend with these structures is understanding the length scale of turbulence encountered by fishes and whether or not fish prefer and could even benefit from a turbulent environment [61]. Our experiments utilized a passive turbulence grid to generate turbulent eddies with a length scale approximating the body depth of the giant danio studied. In nature, however, turbulent flows can differ in the turbulence length scale, in the energy present at each eddy size, and variation in turbulence scale can be important for understanding the effects of turbulence on animals of different sizes. It remains unknown if fish (either individuals or schools) select particular turbulent length scales when swimming that could also allow locomotor energy savings in contrast to the increased costs demonstrated at other length scales. Perhaps the design of fish passage devices for habitat restoration should consider the ratio of turbulence eddy scale relative to the animal size to improve locomotor ability and reduce the cost of movement for fish [58]. Alternatively, energetically costly turbulence flow generators could serve as aquatic barriers to perturb invasive species.

Given the ubiquity of turbulent flows in natural aquatic ecosystems, we suggest that one of the important roles of collective behavior in fish species is to shelter individuals within a collective group from challenging hydrodynamic conditions. More broadly, our study proposes that vertebrate collectives can also function as a larger size biological entity, which could reduce the effect of turbulent perturbation on animal movement. Using aquatic vertebrates moving in the dense water fluid as a model system to directly demonstrate energy saving can be the foundation for future studies of the "turbulence sheltering" hypothesis in flying and terrestrial vertebrates. Locomotor performance curves, where metabolic or kinematic variables are evaluated against swimming speed, are a useful comparative framework to broadly understand the energetic cost of collective movement [34].

## Materials and methods

### Experimental animals

Experiments were performed on giant danio (*D. aequipinnatus*) that were acquired from a local commercial supplier near Boston, Massachusetts, United States of America. Five schooling groups are randomly distributed and housed separately in five 37.9 l aquaria (*n* = 8 per tank). The 5 solitary individuals are housed separately in five 9.5 l aquaria (*n* = 1 per tank). All aquaria have self-contained thermal control (28˚C), an aeration system (>95% air saturation, % sat.) and a filtration system. Water changes (up to 50% exchange ratio) were carried out

weekly. Fish were fed ad libitum daily (TetraMin, Germany). Animal holding and experimental procedures were approved by the Harvard Animal Care IACUC Committee (protocol number 20-03-3).

## Experimental system—Integrated Biomechanics & Bioenergetic Assessment System (IBAS)

The experimental system and similar experimental protocols are available in [62]. To promote reproducibility, we reiterate the methodologies in detail and the additional experimental detail specific to the study of collective movement in turbulent conditions.

The core of our experimental system is a 9.35-l (respirometry volume plus tubing) customized Loligo swim-tunnel respirometer (Tjele, Denmark). The respirometer has an electric motor, and a sealed shaft attached to a propeller located inside the respirometer. By regulating the revolutions per minute (RPM) of the motor, the water velocity of the motor can be controlled.

The swim-tunnel respirometer is oval-shaped. The central hollow space of the respirometry increases the turning radius of the water current. As a result, the water velocity passing the cross-section of the swimming section ($80 \times 80 \times 225$ mm) is more homogenous (validated by PIV). Moreover, a honeycomb flow straightener ($80 \times 80 \times 145$ mm) is installed in upstream of the swimming section to create laminar flow (validated by PIV). The linear regression equation between RPM and water velocity (V) of laminar flow is established (V = 0.06169•RPM– 5.128, $R^2$ = 0.9988, $p < 0.0001$) by velocity field measured by particle image velocimetry (PIV).

To increase the signal-to-noise ratio for the measurement of water dissolved $O_2$, a water homogenous loop is installed 95 cm downstream of the propeller and the water is returned to the respirometer 240 mm before the swimming section. The flow in the water homogenous loop moves (designated in-line circulation pump, Universal 600, EHEIM GmbH & Co KG, Deizisau, Germany) in the same direction as the water flow in the swimming tunnel. A high-resolution fiber optic $O_2$ probe (Robust oxygen probe OXROB2, PyroScience GmbH, Aachen, Germany) is sealed in the homogenous loop at downstream of the circulation pump (better mixing) to continuously measure the dissolved $O_2$ level in the water (recording frequency approximately 1 Hz, response time <15 s). The oxygen probe was calibrated to anoxic (0% sat., a solution created by super-saturated sodium sulphite and bubbling nitrogen gas) and fully aerated water (100% sat.). The background $\dot{M}O_2$ in the swim-tunnel respirometer was measured for 20 min before and after each trial. The average background $\dot{M}O_2$ (<6% of fish $\dot{M}O_2$) was used to correct for the $\dot{M}O_2$ of fish. The prefiltered water (laboratory grade filtration system) is constantly disinfected by UV light (JUP-01, SunSun, China) located in an external water reservoir to suppress bacterial growth. Water changes of 60% total volume occurred every other day and a complete disinfection by sodium hypochlorite is conducted weekly (Performance bleach, Clorox and 1,000 ppm).

To simultaneously measure schooling dynamics and swimming kinematics, the customized oval-shaped swim-tunnel respirometer is located on a platform with an open window beneath the swimming section. The platform is elevated 243 mm above the base to allow a front surface mirror to be installed at a 45˚ angle. This mirror allows a high-speed camera (FASTCAM Mini AX50 type 170K-M-16GB, Photron Inc., United States, lens: Nikon 50mm F1.2, Japan) to record the ventral view. The second camera (FASTCAM Mini AX50 type 170K-M-16GB, Photron Inc., United States, lens: Nikon 50mm F1.2, Japan) is positioned 515 mm to the side of the swimming section to record a lateral view. Synchronized lateral and ventral video recordings were made at 125 fps, and each frame was 1,024 by 1,024 pixels. To avoid light refraction passing through the water and distorting the video recordings, the swim-tunnel respirometry

is not submerged in the water bath. Temperature regulation of the respirometer is achieved by regulating room temperature, installing thermal insulation layers on the respirometry and replenishing the water inside the respirometer from a thermally regulated (28°C, heater: ETH 300, Hydor, United States and chiller: AL-160, Baoshishan, China) water reservoir (insulated 37.9-l aquarium) located externally.

The aerated (100% sat., air pump: whisper AP 300, Tetra, China) reservoir water is flushed (pump: Universal 2400, EHEIM GmbH & Co KG, Deizisau, Germany) to the respirometer through an in-line computer-controlled motorized ball valve (US Solid) installed at the in-flow tube. The other in-line one-way valve is installed at the out-flow tube. The out-flow tube is also equipped with a valve. The value is shut during the measurement period, a precautionary practice to eliminate the exchange of water between the respirometer and the external reservoir when the water moves at a high velocity inside the respirometer. This flushing was manually controlled to maintain DO above 80% sat. Every time the respirometer was closed to measure $\dot{M}O_2$, the water temperature fluctuates no more than 0.2°C. The water temperature inside the respirometer is measured by a needle temperature probe (Shielded dipping probe, PyroScience GmbH, Aachen, Germany) sealed through a tight rubber port of the respirometer.

To allow fish to reach the undisturbed quiescent state during the trial, the entire Integrated Biomechanics & Bioenergetic Assessment Platform (IBAP) is covered by laser blackout sheets (Nylon Fabric with Polyurethane Coating; Thorlabs Inc., New Jersey, United States). The room lights are shut off and foot traffic around the experimental rig is restrained to the absolute minimum. Fish are orientated by dual small anterior spots of white light (lowest light intensity, Model 1177, Cambridge Instruments Inc., New York, United States) for orientation (one to the top and the other to the side) of the swimming section. The test section is illuminated by infrared light arrays.

## Creating turbulent flows in swim-tunnel respirometer

We used a passive turbulence grid (height × width: 7.5 × 8.6 cm) to generate the turbulence flow for the swimming section in the swim-tunnel respirometer (Fig A in S1 Text). The turbulence grid has a configuration of 3 × 3 square openings (each opening is 1.5 × 1.5 cm). The openings produce 9 streams of jets which mix and form turbulent flow. The turbulent grid is upstream of the swimming section (Fig A in S1 Text). The opening of the grid is guarded by thin metal wires to prevent fish from going through. The turbulence grid is effective in generating turbulence, as illustrated by the fluid dynamic features of the turbulences measured by PIV (see Figs B and C in S1 Text). As a result, the linear regression equation between RPM and average water velocity (V) of turbulent flow is changed (V = 0.03515•RPM− 1.597, $R^2$ = 0.9985, $p < 0.0001$) and quantified by velocity field measured by PIV (see Fig C in S1 Text). Quantifying average water velocity allowed us to match the swimming kinematics and metabolic energy consumption of tested fish at the same mean speed between laminar and turbulent flows.

## Experimental protocol

The same individuals or schools are repeatedly measured in laminar or turbulent flow to control for biological variations. Giant danio (*D. aequipinnatus*) is a model species, capable of actively and directionally swimming from a minimum to maximum sustained speeds (0.3 to 8.0 body lengths $s^{-1}$; BL $s^{-1}$ and Reynolds number range of 6.4•$10^3$ to 1.8•$10^5$ in laminar flow). We studied 5 replicate schools and 5 replicate individuals drawn from within each school. Swimming performance test trials were conducted with *D. aequipinnatus* fasted for 24 h, a sufficient period for a small-sized species at 28°C (i.e., high resting $\dot{M}O_2$) to reach an absorptive

state. In fact, we observed no specific dynamic action, in the amount of oxygen consumed for digestion during the first diurnal cycle (Fig C in S1 Text). Prior to the swimming performance test, testing fish were gently weighted and placed in the swim-tunnel respirometer. The fish swam at 35% $U_{crit}$ for 30 min to help oxidize the inevitable but minor lactate accumulation during the prior handling and help fish become accustomed to the flow conditions in the swim-tunnel respirometer [63]. After this time, the fish to be tested were habituated (>20 h) to the respirometer environment under quiescent and undisturbed conditions. During this time, we used an automatic system to measure the resting $\dot{M}O_2$ for at least 19 h. Relays (Cleware GmbH, Schleswig, Germany) and software (AquaResp v.3, Denmark) were used to control the intermittent flushing of the respirometer with fresh water throughout the trial to ensure $O_2$ saturation of the respirometer water. $\dot{M}O_2$ was calculated from the continuously recorded dissolved $O_2$ level (at 1 Hz) inside the respirometer chamber. The intermittent flow of water into the respirometer occurred over 930 s cycles with 30 s where water was flushed into the respirometer and 900 s where the pumps were off and the respirometer was a closed system. The first 240 s after each time the flushing pump was turned off were not used to measure $\dot{M}O_2$ to allow $O_2$ levels inside the respirometer to stabilize. The remaining 660 s when the pumps were off during the cycle were used to measure $\dot{M}O_2$. The in-line circulation pump for water in the $O_2$ measurement loop stayed on throughout the trial.

We characterize the locomotor performance curve of fish using an established incremental step-wise critical swimming speed ($U_{crit}$) test [35]. The first preliminary trial determined the $U_{crit}$ of this population of *D. aequipinnatus* as 8 BL s$^{-1}$. Characterizing the swimming performance curve required a second preliminary trial to strategically select 10 water velocities (0.3, 0.5, 0.8, 1.0, 1.3, 1.5, 1.8, 2.3, 2.8 BL s$^{-1}$) to bracket the hypothesized concave upward metabolism-speed curve at the lower speed (<40% $U_{crit}$). Additional 5 water velocities (3.8, 4.9, 5.9, 6.9, and 8.0 BL s$^{-1}$) are used to characterize the exponentially increasing curve to the maximum and sustained swimming speed, $U_{crit}$ (see Fig E in S1 Text). Altogether, 14 points provide a reliable resolution to characterize the locomotor performance curve. At each water velocity, fish swam and exercised for 10 min [64,65] to reach a steady state in $\dot{M}O_2$ at low speeds (see Fig F in S1 Text). Above 40% $U_{crit}$, $\dot{M}O_2$ can become more variable [66]. Hence, in this protocol, we focus on measuring the sustained aerobic energy expenditure by calculating the average $\dot{M}O_2$ for each 10-min velocity step using Eq 1. The respirometry system reaches a stable signal-to-noise ratio once the sampling window is longer than 1.67 min (see Fig G in S1 Text), well within the duration of the velocity step to obtain a stable signal-to-noise ratio for calculating $\dot{M}O_2$ [66]. At the fifth min of each velocity step, both ventral and lateral-view cameras are triggered simultaneously to record 10-s footage at 125 frames per second, at 1/1,000 shutter speed and 1,024 × 1,024 pixel resolution. Thus, both data streams of $\dot{M}O_2$ and high-speed videos are recorded simultaneously. The $U_{crit}$ test is terminated when 12.5% of fish in the school or a solitary individual touches the back grid of the swimming section for more than 20 s [63]. The $U_{crit}$ test lasted approximately 140 min and estimates the aerobic portion of energy expenditure over the entire range of swimming performance.

To measure the contribution of non-aerobic $O_2$ cost, where most of the cost is related to substrate-level phosphorylation, and to calculate the TEE for swimming over the entire speed range, we measured EPOC after the $U_{crit}$ test for the ensuing 19 h, recorded by an automatic system. Most previous measurements of EPOC after $U_{crit}$ test have used a duration of approximately 5 h, but our extended measurement period ensured that longer duration recovery $O_2$ consumption (EPOC) was measured completely as fish were exercised to $U_{crit}$ (see summary table in 16). The intermittent flow of water into the respirometer occurred over 30 s to replenish the dissolved $O_2$ level to approximately 95% sat. For the following 900 s the flushing pump

remained closed, and the respirometer became a closed system, with the first 240 s to allow $O_2$ saturation inside the respirometer to stabilize. The remaining 660 s when the flushing pump was off during the cycle were used to measure $\dot{M}O_2$ (see Eq 1). The cycle is automated by computer software (AquaResp v.3) and provided 74 measurements of $\dot{M}O_2$ to compute EPOC. Upon the completion of the three-day protocol, the school or individual fish are returned to the home aquarium for recovery. The fish condition was closely monitored during the first 48 h after the experiment, during which no mortality was observed.

## Bioenergetic measurement and modeling

To estimate the steady-rate whole-animal aerobic metabolic rate, $\dot{M}O_2$ values were calculated from the sequential interval regression algorithm (Eq 1) using the dissolved $O_2$ (DO) points continuously sampled (approximately 1 Hz) from the respirometer.

$$\dot{M}O_2 = \left[ \frac{d_{DO[i,(i+a)]}}{d_{t[i,(i+a)]}} \bullet (V_r - V_f) \bullet S_O \right] / (t \bullet M_f) \qquad \text{(Eq 1)}$$

Where $d_{DO}/d_t$ is the change in $O_2$ saturation with time, $V_r$ is the respirometer volume, $V_f$ is the fish volume (1 g body mass = 1 ml water), $S_o$ is the water solubility of $O_2$ (calculated by AquaResp v.3 software) at the experimental temperature, salinity, and atmospheric pressure, $t$ is a time constant of 3,600 s h$^{-1}$, $M_f$ is fish mass, and $a$ is the sampling window duration, $i$ is the next $PO_2$ sample after the preceding sampling window. $\dot{M}O_2$ of the low data quality (e.g., nonlinear decline or $R^2 < 0.5$) are removed to control for the measurement errors [36,67–69].

To account for allometric scaling, the $\dot{M}O_2$ values of solitary fish were transformed to match the size of the individual fish in the school using an allometric scaling exponent (b = 0.7546). The calculation of the scaling relationship [$Log_{10}(\dot{M}O_2) = b \bullet Log10(M) + Log10(a)$, where $M$ is the body mass and $a$ is a constant] was performed by least squares linear regression analysis (y = 0.7546•× + 0.2046; $R^2$ = 0.6727, $p < 0.0001$) on the 180 data points of metabolic rate and body mass from a closely related species (the best available data set to our knowledge) [70]. The allometrically scaled $\dot{M}O_2$ values were used to derive other energetic metrics (listed below) for the solitary fish. The energetic metrics of fish schools are calculated from the mass-specific $\dot{M}O_2$.

The resting oxygen uptake ($\dot{M}O_{2rest}$), the minimum resting metabolic demands of a group of fish or a solitary individual, is calculated from a quantile 20% algorithm [68] using the $\dot{M}O_2$ estimated between the 10th–18th hour and beyond the 32nd hour of the trial. These are the periods of quiescent state when fish completed the EPOC from handling and swimming test.

The EPOC is an integral area of $\dot{M}O_2$ measured during post-exercise recovery, from the end of $U_{crit}$ until reached $\dot{M}O_{2rest}$ plus 10% [36]. This approach reduces the likelihood of overestimating EPOC due to spontaneous activities [36] while taking into account a conceptual framework to estimate the anaerobic costs for activities [71–74]. To account for the allometric scaling effect, we used the total amount of $O_2$ consumed (mg $O_2$) by the standardized body mass of fish (1.66 g) for fish schools and solitary fish.

We model EPOC (i.e., non-aerobic $O_2$ cost) to estimate a total $O_2$ cost over the duration of the swimming performance test. Our conceptual approach was pioneered by Brett [35] in fish and is also used in sports science [33]. Mathematical modeling was applied to study the effects of temperature on the cost of swimming for migratory salmon [63]. We improved the mathematical modeling by applying the following physiological and physics criteria. The first criterion is that significant accumulation of glycolytic end-product occurred when fish swimming

above 50% $U_{\text{crit}}$ [75] which corresponds to > approximately 40% $\dot{M}O_{2\text{max}}$ (or approximately 50% aerobic scope) [33]. This is also when fish start unsteady-state burst-&-glide swimming gait [75]. The second criterion is that the integral area for the non-aerobic $O_2$ cost during swimming can only differ by $\leq 0.2\%$ when compared to EPOC. The non-aerobic $O_2$ cost during swimming is the area bounded by modeled $\dot{M}O_2$ and measured $\dot{M}O_2$ as a function of time when fish swim >50% $U_{\text{crit}}$ (see Fig 2A and S2 Table in S1 Text). The third criterion is that TEE is expected to increase exponentially with swimming speed (Fig G in S1 Text). Specifically, these curves were fitted by power series or polynomial models, the same models that describe the relationship between water velocity and total power and energy cost of transport (Fig G in S1 Text). Following these criteria, the non-aerobic $O_2$ cost at each swimming speed is computed by a percentage (%) modifier based on the aerobic $O_2$ cost (S1 and S2 Tables in S1 Text). The exponential curve of total $O_2$ cost as swimming speed of each fish school or solitary individual was derived by an iterative process until the difference between non-aerobic $O_2$ cost and EPOC met the second criterion. The sum of non-aerobic $O_2$ cost and aerobic cost gives the total $O_2$ cost.

The best model fitting following the relationships between water velocity and energetic costs of locomotion suggests that the glycolysis starts at 2 to 3 BL s$^{-1}$ for fish swimming in the turbulence flow, whereas the same model suggests that the glycolysis starts at 4 to 5 BL s$^{-1}$ for fish swimming in the laminar flow. We are confident in this model, because the engaging glycolysis at the lower swimming speed for the fish swimming in turbulent flow corresponded with their lower $U_{\text{crit}}$ (laminar flow: 8 BL s$^{-1}$ versus turbulent flow: 7 BL s$^{-1}$). The model of aerobic and non-aerobic energy costs of locomotion enabled the estimation of total energy expenditure and total cost of transport as detailed below:

TEE is calculated by converting total $O_2$ cost to kJ $\bullet$ kg$^{-1}$ using an oxy-calorific equivalent of 3.25 cal per 1 mg $O_2$ [76].

Total cost of transport (COT), in kJ $\bullet$ km$^{-1}$ $\bullet$ kg$^{-1}$ is calculated by dividing TEE by speed (in km $\times$ h$^{-1}$) [64].

## Hydrodynamic flow visualization and analysis

We used a horizontal plane of laser sheet (LD pumped all-solid-state 532 nm green laser, 5w, MGL-N-532A, Opto Engine LLC) to visualize and quantify the hydrodynamic conditions of laminar and turbulent flows (i.e., PIV). The horizontal planes of the water fluid field in laminar and turbulent conditions were calculated from consecutive video frames (1,024 × 1,024 pixels) using DaVis v8.3.1 (LaVision Inc., Göttingen, Germany). A vector field (covering a horizontal plane of 34.3 cm$^2$ with 2,193 vectors) is characterized by a sequential cross-correlation algorithm applied with an initial interrogation window size of 64 × 64 pixels that ended at 12 × 12 pixels (3 passes, overlap 50%). To optimize the signal-to-noise ratio on the sequential cross-correlation algorithm, we used an increment of 10 frames at the lowest speed of laminar and turbulent flow conditions. The rest of the speeds in both flow conditions were analyzed using an increment of 1 frame. Five sets of consecutive video frames (1st, 250th, 500th, 750th, and 990th frame out of 1,000 frames) in laminar ($n = 5$) and turbulent flows ($n = 5$) were used for calculating the metrics.

To optimize the sampling resolution of consecutive PIV frames, the six lowest testing speeds (106, 131, 156, 181, 256, and 356 RPM) of both flow conditions were captured by 1,000 frame rate per sec (shutter speed: 1/1,000), whereas the 6 highest testing speed (456, 556, 656, 756, 856 RPM) of both flow conditions were captured by 2,000 frame rate per sec (shutter speed: 1/1,000) using a high-speed camera (FASTCAM Mini AX50 type 170K-M-16GB, Photron Inc., United States, lens: Nikon 50 mm F1.2, Japan).

From the fluid field, we used DaVis (v8.3.1) to calculate the following fluid parameters:

$V_{max}$ (m s$^{-1}$): Extract the maximum value of the velocity at the vector that is perpendicular to the free-stream flow.

Maximum vorticity (sec$^{-1}$): calculated according to the central difference scheme with 4 closest neighbors at the horizontal plane. This method achieves a high spatial resolution.

Maximum shear strength (1/S$^2$): maximum value of the shear strength over the time series of flow images analyzed. Shear strength is the positive eigenvalue of the matrix for two-dimensional vorticity on the horizontal plane.

Sum of shear strength (1/S$^2$): sum of the shear strength in the unit of 1/S$^2$ (positive values) summed over the entire set of video frames to provide another metric of shear induced by the passive turbulence grid. The calculation of shear strength is stated above.

To compute the fluctuation velocity, turbulence intensity, and eddy size distribution in the turbulent flow, a subsection of the full velocity field was extracted. This subsection spans an area of 22.4 cm$^2$ and was located approximately 4 cm from the passive grid (Fig A in S1 Text) and 0.5 cm from the walls of the tunnel. For this analysis, velocity fields were computed on all 2,000 frames. For further information about the quantification of turbulence and the distribution of eddy sizes, readers can refer to the section of computing fluctuation velocity and turbulence intensity (S1 Text).

### Three-dimensional kinematic data extraction from high-speed videography

We used two synchronized 10-s high-speed videos (lateral and ventral views, at each speed) for kinematic analyses. We calibrated the field of view of the high-speed cameras using a direct linear transformation for three-dimensional kinematic reconstruction (DLTdv8) [77] by applying a stereo calibration to the swimming section of the respirometer (see Fig H in S1 Text). We digitized the anatomical landmarks of fish (nostrils) to obtain the X, Y, Z coordinates for each marker at the 1st s, 5th s, and 10th s for videos recorded at each speed. These coordinates are used to calculate the following kinematic parameters. All the calculations are validated on the known length and angle of test objects inserted into the tank working section.

Strouhal number (St) represents the dimensionless flapping frequency and amplitude at a given movement speed. St $= \frac{fA}{U}$, where $f$, $A$ and $U$ are the tailbeat frequency, amplitude, and swimming speed [13], respectively. The measurement is conducted on the calibrated high-speed video in video analysis software (Photron FASTCAM Viewer 4, Photron USA).

Reynolds number (Re) represents the dimensionless fluid inertial to viscous forces at a given speed. Re $= \frac{\rho U L}{\mu}$, where $\rho$ and $\mu$ are the density and dynamic viscosity of the water, and $U$ and $L$ are the swimming speed and length of the fish [13]. Water density and dynamic viscosity are given at 28°C.

In addition to the manual digitization, we also developed a contrast-based segmentation algorithm for automatic tracking of 2D kinematics in the ventral view (see Harvard Dataverse: doi.org/10.7910/DVN/CVNLZE). The video analysis was performed in MATLAB (R2022b). We processed each frame in the video independently. We first converted the frame into grayscale and then used a brightness threshold to obtain masks of the fish (panel (A) of Fig I in S1 Text). We removed masks with an area larger than that expected for a single fish, excluding masks with multiple overlapping fish. For every streamwise location on the mask (x-location), we calculated the midpoint across the span to obtain the midline of the animal at each instant. The midlines were pieced together across frames according to their locations. We centered and rotated a series of midlines to account for the rigid-body component of the fish body. At this stage of the image analysis, we obtained midline envelopes of the fish (panel (B) Fig I in S1 Text). We extracted a time series of head and nose spanwise oscillation, identified peaks and troughs of the signal and calculated the amplitude and frequency of the fish (panels (C) and (D)

Fig I in S1 Text). We excluded time series that are shorter than a tail beat cycle and those associated with unsteady swimming, either with a high relative velocity with flow or fast rotation.

To further characterize the head oscillation of fish in different flow environments, we processed trajectories that are several dozen cycles long at 6 BL s$^{-1}$, extracted through manual digitization (DLTdv8a) (8.2.10) [76]. To distinguish the nose oscillation from the background movement of the fish during the period, we performed fast Fourier transform (FFT) in Lab-Chart v7.3.8. The FFT analyses enabled us to separate the original time series of the nose trajectories into background motion (low-frequency component) and oscillation due to fish swimming (high-frequency component). The latter is plotted in Fig 7A. The power spectrum analysis is conducted in LabChart (v.7.2.8) where power spectral density is calculated as squared mean amplitude at each frequency.

## Statistical analyses

Measurement points are presented as mean ± SEM. For the metrics that failed normality tests, logarithm transformations were applied to meet the assumptions of normality of residuals, homoscedasticity of the residuals, and no trend in the explanatory variables. We conducted supervised statistical tests to specifically evaluate our hypotheses about the effects of turbulence on the biomechanics and bioenergetics of fish swimming, either in school or alone. The statistical comparisons for the different responses of fish schools (or solitary fish) between swimming in laminar flows and in turbulent flow used a mixed effects model (laminar flow versus turbulent flow and swimming speed) with Holm–Šídák post hoc tests. The statistical comparisons for the difference between fish schools and solitary fish swimming in turbulence used a general linear model (solitary fish versus fish schools and swimming speed) with Holm–Šídák post hoc tests. The statistical comparison for the characteristics of fluid dynamics between laminar and turbulent flow conditions used a general linear model that used speed as a covariance. The statistical comparison of EPOC between fish schools and solitary fish after performing the $U_{crit}$ test in turbulent conditions used an unpaired $t$ test. The statistical analyses were conducted in SPSS v.28 (SPSS Inc. Chicago, Illinois, USA). The best-fitting regression analyses were conducted using Prism v.9.4.1 (GraphPad Software, San Diego, California, USA), and 95% CI values were presented for all regression models as shaded areas around the regression or data points. Statistical significance is denoted by *, **, ***, **** for $p$-values of $\leq 0.05$, $\leq 0.01$, $\leq 0.001$, $\leq 0.0001$, respectively.

## Supporting information

**S1 Text. Supplementary materials for collective movement of schooling fish reduces the costs of locomotion in turbulent conditions. Fig A. Photograph of the passive turbulence grid used to generate the turbulent flows for the fish swimming section inside the swim-tunnel respirometer.** Each opening is 1.5 × 1.5 cm. The plate has a height and width of 7.5 × 8.6 cm. The turbulence grid is inserted in front of the fish swimming section. The pass of the laminar-like flow through the grid generates higher velocity jets through the openings. The jets of flow then mix and create a turbulent environment for the entire swimming section (see S2 Fig for the characterizations of the turbulence in S1 Text). **Fig B. Characteristics of the turbulence generated by a passive grid for different mean velocities.** (A) Transverse ($D_{NN}$) and longitudinal ($D_{LL}$) structure functions for 2, 4, 6, 8 $BLs^{-1}$. In the inertial subrange, the structure functions scale as $r^{2/3}$ which agrees with Kolmogorov's local isotropy hypothesis. (B) Energy dissipation rate is computed from $D_{NN}$ and $D_{LL}$ using the relationship shown. The energy dissipation rate was approximated as the average maximum value between the 2 curves. (C) One-dimensional energy spectrum for each mean flow tested. The x-axis denotes the wavenumber

($K$) normalized by the integral length scale. The y-axis denotes the amount of energy contained in eddies of a certain wavenumber. The energy is normalized by the integral length scale times the square of the total fluctuation velocity. The dashed line demonstrates that the energy scales as the wavenumber to the power minus five-thirds, agreeing with Kolmogorov's hypothesis. The underlying data of this figure are in doi.org/10.7910/DVN/CVNLZE. **Fig C. The mathematical relationship between average water velocity and revolutions per minute in the swim-tunnel respirometer.** The average water velocity (denoted as "y") is measured at the swimming section of the respirometer by particle image velocimetry (PIV). The revolutions per minute (RPM, denoted as "x") is the rotation speed of the motor on the swim-tunnel respirometer. The average water velocity in the swimming section is quantified in both laminar (green) and turbulent (orange) flow conditions. The linear regression equations are stated in the figure legends. The underlying data of this figure are in doi.org/10.7910/DVN/CVNLZE. **Fig D. A schematic illustration of the incremental steps in swimming speed used for the critical swimming speed test ($U_{crit}$) protocol.** Speed is presented as the relative swimming speed of fish normalized to body lengths per second (BL s$^{-1}$). Each speed increment has a 10-min duration. Each vertical increment marks the increase in speed for each step. The mean velocity at fatigue is indicated by the * symbol, marking the total duration of the test. The methodological details are adapted from [62] Zhang and Lauder (2024), eLife, 12, doi:10.7554/eLife.90352.2 as both studies used the same experimental system and similar experimental protocols. **Fig E. Stability of whole-animal oxygen uptake ($\dot{M}O_2$) profile when fish swim steadily at low water velocity**. The quality of $\dot{M}O_2$ measurement over the range of relative swimming speeds (body length per sec, BL s$^{-1}$) where fish show the lowest $\dot{M}O_2$ is critical to assure the accuracy of the $J$-shaped metabolism-speed curve. Hence, we conducted an additional quality assurance test to inspect the stability of $\dot{M}O_2$ at 1.25 (A), 1.5 (B), 1.75 (C), and 2.0 (D) BL s$^{-1}$ over a 30-min period. $\dot{M}O_2$ recorded in the first 10 min is within the same range as the ensuing 20 min. Thus, our 10-min testing period at each water velocity provides a reliable estimate of the aerobic cost when fish swim at a given swimming speed. This conclusion is in agreement with the same testing period used in a previous study ([64] V. Di Santo and colleagues, 2017 PNAS 114, 13048–13053). The methodological details are adapted from [62] Zhang and Lauder (2024), elife, 12, doi:10.7554/eLife.90352.2 as both studies used the same experimental system and similar experimental protocols. **Fig F. An analysis of the impact of varying the sampling window duration on estimates of individual rates of oxygen uptake ($\dot{M}O_2$).** The signal-to-noise ratio analyses used 6 independent background respirometer $\dot{M}O_2$ data sets in a 20-min duration. We calculated (A) standard deviations (SD) and (B) coefficient of variation (CV) for each complete data set as a function of sampling window duration (0.8 to 5.0 min). The SD and CV values were compared across different sampling window durations using one-way ANOVA with Tukey post hoc tests ($\alpha < 0.05$). Values are presented as mean ± SEM. This analysis suggests that 1.67-min as a minimum conservative sampling duration (the red vertical dashed line) using the criterion of SD and CV being stable after high variation at shorter duration windows. This suggests that the 8-min sampling duration used in this study is more than sufficient to resolve the measurement of steady-state $\dot{M}O_2$ for sustained swimming. The methodological details are adapted from [62] Zhang and Lauder (2024), elife, 12, doi:10.7554/eLife.90352.2 as both studies used the same experimental system and similar experimental protocols. **Fig G. Illustration of the theoretical relationship between the cost of transport and total power as a function of nominal speed based on the estimated power required to swim as drag forces increase at increasing swimming speeds.** The equation for the energy cost of transport is $y = 0.9996x^{-1} + x^2$ (power series model: $R^2 = 1$, AIC = −127.6). The equation of total power is $y = 1.001 - 0.000643x + 8.564e^{-005}x^2 + 1x^3$ (polynomial model:

$R^2 = 1$, AIC = −138.7). Notably, the same models provided best-fit equations to describe the measurement of the cost of transport and total energy expenditure (aerobic and glycolytic metabolism) on live fish. This suggests that total energy expenditure during fish locomotion at increasing speed may be largely due to the requirement to overcome fluid dynamic drag. The theoretical numbers (nominal) of power and energy cost of transport are adopted from D. Robinson, Ed., Animal Performance (The Open University, Milton Keynes, 1997). Speed is given in body lengths per second (BL $s^{-1}$) while the power and energy axes are unitless relative numbers. The methodological details are adapted from [62] Zhang and Lauder (2024), eLife, 12, doi:10.7554/eLife.90352.2 as both studies used the same experimental system and similar experimental protocols. **Fig H. Photographs of the "spike" calibration device used to map the three-dimensional space of the fish swimming section inside the swim-tunnel respirometer.** This direct linear transformation calibration system provided accurate 3D coordinates within the swimming arena for both individual danio and schools of up to 8 danio. The tip of each spike provides a marker for spatial coordinates. Each tip has a known three-dimensional coordinate based on the known distance on the x-(length), y-(width), and z-(height) axes. Lateral (A) and dorsal (B) views are provided. The "spike" calibration provides 45 spatial points to sufficiently cover a volume of $1440 \cdot 10^3$ $mm^3$. The spatial calibration is used by DLTdv8 software (Hedrick (2008), Bioinspiration & Biomimetics 3, 034001) to extract the three-dimensional coordinates for locations digitized on individual fish (see S9 Fig). The methodological details are adapted from [62] Zhang and Lauder (2024), eLife, 12, doi:10.7554/eLife.90352.2 as both studies used the same experimental system and similar experimental protocols. Fig I. Extractions of individual fish kinematics in the schools. (A) A representative video frame converted into black and white. The region and the midline representing the fish being tracked are highlighted. (B) Midlines are obtained by processing consecutive video frames. Time series of (C) head and (D) tail oscillations in periodic patterns. The peaks and troughs are marked in black circles. Table A. Parameters for the metabolic model for calculating the total $O_2$ cost during a critical swimming speed ($U_{crit}$) test in turbulent conditions. This model adds the amount of $O_2$ consumed post $U_{crit}$ test (excess post-exercise $O_2$ consumption, EPOC, calculated by an area under the curve algorithm, AUC) on the active $O_2$ uptake ($\dot{M}O_2$) over 2–7 body lengths per second (BL $s^{-1}$) when fish use $\geq$20% maximum $\dot{M}O_2$ ($\dot{M}O_{2max}$). A threshold level of approximately 40% $\dot{M}O_{2max}$ is the workload that initiates glycolytic metabolism in fish schools, whereas the modeling indicated that the glycolytic metabolism is initiated at approximately 27% $\dot{M}O_{2max}$ in solitary individuals. Glycolytic metabolic cost is a major contributor to EPOC in addition to the use of high-energy phosphates. The model uses a percentage (%) modifier to compute the $O_2$ cost in addition to the active $\dot{M}O_2$ at each swimming speed. The key criterion is that the area of non-aerobic $O_2$ cost above the $\dot{M}O_2$ should be equal to EPOC, as shown in the section of the table as EPOC validation. The added area of non-aerobic $O_2$ cost is calculated by the delta of AUC for the measured $\dot{M}O_2$ and the AUC for the $\dot{M}O_2$ model. As a result, the calculated total energy expenditure is modeled by the same equation for the theoretical relationship between total power and swimming speed (S7 Fig). Modeling is performed on each school and each solitary individual to calculate the 95% CI and the group average is presented in Fig 4. The modeling for the aerobic and anaerobic metabolic contribution of *D. aequipinnatus* to swim in the laminar flows is available in [62] Zhang and Lauder (2024), eLife, 12, doi:10.7554/eLife.90352.2. Table B. Energetic contributions from aerobic and glycolytic metabolism for swimming in turbulent conditions. The partitions of metabolic contribution (mean ± SEM) are detailed for giant danio (*D. aequipinnatus*) to swim at each testing speed that engages the glycolytic metabolism. The values are calculated from the metabolic modeling (Table S1). The partitions of aerobic and anaerobic metabolic contribution for *D.*

*aequipinnatus* to swim in the laminar flows are available in [62] Zhang and Lauder (2024), eLife, 12, doi:10.7554/eLife.90352.2.
(DOCX)

## Acknowledgments

Many thanks to members of Lauder Laboratory for numerous discussions about fish schooling behavior, for comments on the manuscript, to Dr. Robin Thandickal for assistance with the convex hull calculations, and to Cory Hahn for fish care.

## Author Contributions

**Conceptualization:** Yangfan Zhang, Rui Ni, George V. Lauder.

**Funding acquisition:** Yangfan Zhang, Rui Ni, George V. Lauder.

**Investigation:** Yangfan Zhang, Hungtang Ko, Michael A. Calicchia.

**Methodology:** Yangfan Zhang, Hungtang Ko, Michael A. Calicchia, Rui Ni, George V. Lauder.

**Project administration:** Rui Ni, George V. Lauder.

**Supervision:** Rui Ni, George V. Lauder.

**Visualization:** Yangfan Zhang, Hungtang Ko, Michael A. Calicchia, Rui Ni, George V. Lauder.

**Writing – original draft:** Yangfan Zhang.

**Writing – review & editing:** Yangfan Zhang, Hungtang Ko, Michael A. Calicchia, Rui Ni, George V. Lauder.

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
