## [Editor Report · Decision Letter 0]

16 Jan 2024

Dear Dr Zhang, 

Thank you for submitting your manuscript entitled "Collective movement of schooling fish reduces locomotor cost in turbulence" for consideration as a Research Article by PLOS Biology. Please accept my apologies for the delay incurred as we recovered from the holiday period.

Your manuscript has now been evaluated by the PLOS Biology editorial staff, as well as by an academic editor with relevant expertise, and I'm writing to let you know that we would like to send your submission out for external peer review.

Once your full submission is complete, your paper will undergo a series of checks in preparation for peer review. After your manuscript has passed the checks it will be sent out for review. To provide the metadata for your submission, please Login to Editorial Manager (https://www.editorialmanager.com/pbiology) within two working days, i.e. by Jan 18 2024 11:59PM.

Kind regards,

Roli Roberts

Roland Roberts, PhD

Senior Editor

PLOS Biology

rroberts@plos.org

---

## [Decision Letter · Decision Letter 1]

21 Feb 2024

Dear Dr Zhang,

Thank you for your patience while your manuscript "Collective movement of schooling fish reduces locomotor cost in turbulence" went through peer-review at PLOS Biology. Your manuscript has now been evaluated by the PLOS Biology editors, an Academic Editor with relevant expertise, and by three independent reviewers.

You'll see that the reviewers are extremely positive about your study, but each of them has some recommendations for how better to present it in the manuscript. Reviewer #1 thinks that there are some issues with the kinematic analysis, and that the presentation needs substantial improvement (s/he provides a long list of very helpful suggestions). Reviewer #2's suggestions (see the attachment) are mostly presentational, and several involve your treatment of viscosity, turbulence, etc. Reviewer #3 is also very positive, and only has minor requests.

In light of the reviews, which you will find at the end of this email, we are pleased to offer you the opportunity to address the comments from the reviewers in a revision that we anticipate should not take you very long. We will then assess your revised manuscript and your response to the reviewers' comments with our Academic Editor aiming to avoid further rounds of peer-review, although might need to consult with the reviewers, depending on the nature of the revisions.

**IMPORTANT - SUBMITTING YOUR REVISION**

*Resubmission Checklist*

*Published Peer Review*

*PLOS Data Policy*

Sincerely,

Roli Roberts

Roland Roberts, PhD

Senior Editor

PLOS Biology

rroberts@plos.org

REVIEWERS' COMMENTS:

Reviewer #1:

Dear authors,

I believe your paper represents a substantial advance in understanding the benefits of collective locomotion with a set of compelling experiments and high-quality data. I also don't think that the presentation of these results matches the quality of the experiments, reducing the impact of your work and making your own interpretation of your results less clear than it should be. There are also some improvements to be made in the analysis of the kinematic data to better relate them to the energetics.

Analysis: 

First, some comments on tailbeat frequency and amplitude: As you're aware, fluid forces increase with velocity^2. Thus, a simplified model suggests that, all else held equal, a 20% increase in tailbeat frequency would produce a 1.2^2 = 1.44 or 44% increase in force. Furthermore, because mechanical power is given by force * speed and speed increases linearly with frequency, the simple model of expected mechanical power (and muscle power) (and energy expenditure) for that 20% increase in tailbeat frequency is 1.2^2 * 1.2 = 1.2^3 = 1.73. Thus, cost ~ (frequency*amplitude)^3. Note that this model doesn't account for speed from the fish's forward motion but is at least a more physically realistic starting point for your attempt at comparing the kinematic and energetic changes around line 244.

Second, the head oscillations in Figure 7b are an intriguing qualitative look at how schooling affects swimming kinematics, but 1) the "school laminar" condition is missing and 2) the analysis could be made more quantitative by showing the power spectrum instead of just the time series. This should also be presented in the Results, not as an add-on in the Discussion.

Writing and presentation:

Please work to make your presentation of your results consistent, concise, and close to the actual results. For example, in this manuscript you refer to the same flow conditions as "controlled" and "laminar" on different occasions. Pick one! (I prefer laminar). Try and avoid discussion of topics not investigated in your work. For example, in lines 58-62 in only the second paragraph of the paper you digress toward discussing the effect of visual sensing on mitigating turbulence. You results have nothing to say about this and there is little other work to cite, so just go directly from "… alter body kinematics." To the new paragraph beginning with "This challenge…" Please review your text for similar digressions elsewhere.

Don't rely on percentage comparisons, there are lots of ways to calculate them that give somewhat different numbers. PLoS Biology doesn't restrict article length so there's no reason not to provide the actual underlying measurement data.

Specific comments:

Line 27 - collective behavior is a vastly larger topic than locomotion, and many benefits are unrelated to animal movement. Substitute "benefits of collective movement" for "benefits of collective behaviours"

Line 34/35 - It is unclear what "reduced by 63%" means mathematically. English language is very imprecise when describing percentages, so you need to be thorough and provide a bit more context to help the reader decode your meaning. Consider instead "We discovered that, when swimming at high speeds in turbulent flows, fish in a school had only XX-YY% of the total energy expenditure of solitary fish." Alternatively, just provide the measurements, or at least do so along with your percentages.

Line 36 - see prior discussion about "kinematic effort". Also, this line is unrelated to the prior comparison of group effects. Consider introducing the effect of turbulence on solitary fish first, before describing how schooling helps reduce the effect.

Line 37 - "However" suggests you'll provide a follow-up to the prior sentence, but information about school 3D volume is unrelated to measures of individual tailbeat effort. Information about tailbeat effort in schooling fish would be appropriate following the "However", but volume is not.

Line 40 - Didn't you already discuss the effect of schooling versus solitary swimming on TEE back on line 34/35? Why say it again here with a different computation for the percentage result?

Summary: Your existing phrasing obscures your result. Consider "Swimming in a school substantially reduces the energetic cost of movement in turbulent water compared to swimming alone."

Line 61 - This line asserts that animals would have a hard time mitigating the energetic costs of turbulence without ever establishing (by citation or reasoning) that such costs exist. As mentioned before, the 2nd half of this first paragraph should be removed. What remains would flow logically to the second paragraph.

Line 99: you write "whole animal", but your technique seems to measure "whole group" energy expenditure

Line 138: "… energy demands are at a premium." I think I understand your overall point, but this phrase doesn't make sense to me - please rewrite.

Line 146: "1.8-fold faster" Why are you switching between reporting comparisons as percentages and as X-fold differences? Please be consistent in the way you report your results.

Line 149: "62-79% lower" Again, your phrasing doesn't make it immediately obvious what calculation you performed to get your results; please put the actual results in (like you did for EPOC results on line 145) or do a simpler computation, e.g. TEE and TCOT for fish in schools were XX-YY% of solitary fish. "% lower" makes it unclear what the denominator is in your computation.

Line 165 - again, badly reported percentages, better to have the actual numbers

Line 169 - is this result (smaller school size with speed) for turbulent or laminar flow or both?

Line 177 - you use both f and f_tail to mean the same thing. Since the only amplitude discussed here is tailbeat amplitude, perhaps choose a more compact representation than "amp_tail" Greek letter theta would be good if your amplitude is an angle, but whether it is a linear or angular measure isn't clear from the methods.

Line 202: is this supposed to read "less energy than [solitary] fish under control conditions" ? 

Line 202 - This statement is not supported by your results, how do you know that a closely packed school is an essential mechanism? It certainly happened, but you don't have a comparison school that swam in turbulence without close packing to show the benefit of packing. 

Line 204 - Again, you don't provide evidence that this is a result of closer packing

Line 223 - this paragraph re-states the results and should be reduced or removed

Line 235 - another paragraph that restates results instead of discussing them

Line 244 - as noted earlier, your effort = frequency*amplitude model is fluid-dynamically naïve and inadequate to make any comparison with your metabolic measurements

Line 289 - this discussion of oscillation amplitude is interesting and informative, but you should present these Results in the Results section

Line 289 - I'm not sure what you mean here "Individuals within a school swimming in turbulence have rhythmical head oscillations and more distinct peaks than those under turbulent conditions", please rephrase.

Line 293 - if a tighter schooling volume is good and promotes all kinds of energy saving, why don't schools in laminar flow also use the tighter volume? Wouldn't they also like to save energy?

Line 314 - What about CFD simulation? Isn't the Lauder group working with Rajat Mittal on just such a project?

Line 335 - Could you be more clear on how migrating in a group protects against heat, hypoxia and storms? What's your mechanism?

Line 405 - "growth of microbial." Growth of microbial what?

Line 412 & elsewhere: "Phontron" should be "Photron"

Figure 4: it was not at all clear to me that the "model" lines refer to integrated aerobic and non-aerobic costs, please add this information to the figure itself and put it earlier in the caption

Figure 5: the figure legend specifies a comparison of "control" and "laminar", but aren't those the same thing in your work?

Figure 6: please equalize the axis limits between left and right versions of the same measurement type

Reviewer #2:

[IMPORTANT: see attachment for formatted version]

GENERAL

The idea here is that swimming as part of a densely-packed school can lower locomotion costs when the environment is turbulent. A series of experiments were conducted to measure metabolic costs through O2 consumption for solitary and 6-member schools. The authors show quite convincingly that swimming costs are substantially lower, per fish, in schools than alone, and that this difference is much higher when in a turbulent surrounding. The manuscript is very well-written and presented, and will make a notable contribution to the literature. There are a number of comments below that might help improve the clarity and focus of the manuscript.

DETAILS

P4, l63: Yes, viscosity is 50 times that of air. Density is about 1000 times that of air. Is the underlying assumption that the forces to be modified are entirely viscous in nature? Perhaps it would be better here to express the dynamical similarity through the usual Reynolds number.

P4, l113: Correct English: “similar rise of values of all these parameters in control flow condition”

Fig. 1: L/H should be L/D. Why does the shelter envelope stand off from the leading edge of the school? Is the lead fish sheltered?

P4, l107: What is a controlled flow? Is it laminar? (the answer is provided at the end of the caption for Fig. 2, but it ought to be identified in the text too). Can we have some more conventional measures of turbulence for the turbulent and ‘controlled’ flow? In Fig. 2 the contrast flow is termed laminar. Turbulence is often characterized by u’/U where u’ is the fluctuating component of a velocity component, or perhaps of the velocity magnitude. Since fluctuations of course scale with U, then it should be normalized with U also. The top two images in Fig. 2 are not useful. There needs to be a decision here about what quantity to show (the little velocity vectors are mostly invisible). Finally, but most importantly, the quantities need to be defined clearly. What is the coordinate system, what are the velocity components, is this a 2 dimensional system, what are the axes of Fig 2A,B? It might be helpful to express an \\omega as \\omega*D/U where D is a characteristic fish depth (as in Fig. 1). This now compares eddy turnover times to flow advection times. Vorticity *components* need to be clearly identified.

Fig. 3: Some concerns above have been addressed here, and the turbulence intensity (def?) is roughly constant with flow speed, as suspected. Would it make sense now to normalize L with D? Now we have components of velocity in u and v, but it should be made clear again, in the main text or caption, exactly what these are. (Presum

---

## [Editor Report · Decision Letter 2]

12 Apr 2024

Dear Dr Zhang,

Thank you for your patience while we considered your revised manuscript "Collective movement of schooling fish reduces locomotor cost in turbulence" for publication as a Research Article at PLOS Biology. This revised version of your manuscript has been evaluated by the PLOS Biology editors and the Academic Editor.

Based on our Academic Editor's assessment of your revision, we are likely to accept this manuscript for publication, provided you satisfactorily address the following data and other policy-related requests:

IMPORTANT - Please attend to the following:

a) For clarity, for our broader readership, please change your Title slightly to "Collective movement of schooling fish reduces the costs of locomotion in turbulent conditions"

b) Please address my Data Policy requests below; specifically, we need you to supply the numerical values underlying Figs 2ABCDEF, 3ABCD, 4ABCDEFG, 5ABC, 6ABCDEF, 7AB, 8, S2ABC, S3, S4, S5ABCD, S6AB, S7, either as a supplementary data file or as a permanent DOI’d deposition. I note that you already have an associated deposition in Harvard Dataverse, but this is currently not accessible for checking, and is not mentioned in the Data Availability Statement in the manuscript metadata. Please could you ensure that this deposition contains the data and code needed to recreate all of the above-mentioned Figures?

c) Please cite the location of the data clearly in all relevant main and supplementary Figure legends, e.g. “The data underlying this Figure can be found in S1 Data” or “The data underlying this Figure can be found in https://doi.org/10.7910/DVN/CVNLZE"

d) Please make any custom code available, either as a supplementary file or as part of your data deposition.

We expect to receive your revised manuscript within two weeks. 

*Published Peer Review History*

*Press*

Sincerely,

Roli Roberts

Roland Roberts, PhD

Senior Editor

rroberts@plos.org

PLOS Biology

DATA POLICY:

Regardless of the method selected, please ensure that you provide the individual numerical values that underlie the summary data displayed in the following figure panels as they are essential for readers to assess your analysis and to reproduce it: Figs 2ABCDEF, 3ABCD, 4ABCDEFG, 5ABC, 6ABCDEF, 7AB, 8, S2ABC, S3, S4, S5ABCD, S6AB, S7. NOTE: the numerical data provided should include all replicates AND the way in which the plotted mean and errors were derived (it should not present only the mean/average values).

CODE POLICY

Per journal policy, if you have generated any custom code during the curse of this investigation, please make it available without restrictions upon publication. Please ensure that the code is sufficiently well documented and reusable, and that your Data Statement in the Editorial Manager submission system accurately describes where your code can be found.

DATA NOT SHOWN?

---

## [Editor Report · Decision Letter 3]

18 Apr 2024

Dear Dr Zhang,

Thank you for the submission of your revised Research Article "Collective movement of schooling fish reduces the costs of locomotion in turbulent conditions" for publication in PLOS Biology. On behalf of my colleagues and the Academic Editor, Anders Hedenström, I'm pleased to say that we can in principle accept your manuscript for publication, provided you address any remaining formatting and reporting issues. These will be detailed in an email you should receive within 2-3 business days from our colleagues in the journal operations team; no action is required from you until then. Please note that we will not be able to formally accept your manuscript and schedule it for publication until you have completed any requested changes.

Sincerely, 

Roli Roberts

Senior Editor

PLOS Biology

rroberts@plos.org